# Competitive binding of MatP and topoisomerase IV to the MukB hinge domain

**Gemma LM Fisher**[1†‡]**, Jani R Bolla**[2,3†§]**, Karthik V Rajasekar**[1#]**, Jarno Mäkelä**[1¶]**, Rachel Baker**[1]**, Man Zhou**[1**]**, Josh P Prince**[1††]**, Mathew Stracy**[1]**, Carol V Robinson**[2,3]**, Lidia K Arciszewska**[1]**, David J Sherratt**[1*]

[1]Department of Biochemistry, University of Oxford, Oxford, United Kingdom; [2]Physical and Theoretical Chemistry Laboratory, University of Oxford, Oxford, United Kingdom; [3]The Kavli Institute for Nanoscience Discovery, Oxford, United Kingdom

**\*For correspondence:**
david.sherratt@bioch.ox.ac.uk

[†]These authors contributed equally to this work

**Present address:** [‡]Cell Cycle Group, Medical Research Council London Institute of Medical Sciences, London, United Kingdom; [§]Department of Plant Sciences, University of Oxford, Oxford, United Kingdom; [#]Evotec SE, Milton Park, Abingdon, United Kingdom; [¶]ChEM-H Institute, Stanford University, Stanford, United States; [**]Department of Biochemistry, University of Cambridge, Cambridge, United Kingdom; [††]Meiosis Group, Medical Research Council London Institute of Medical Sciences, London, United Kingdom

**Competing interest:** The authors declare that no competing interests exist.

**Abstract** Structural Maintenance of Chromosomes (SMC) complexes have ubiquitous roles in compacting DNA linearly, thereby promoting chromosome organization-segregation. Interaction between the *Escherichia coli* SMC complex, MukBEF, and *matS*-bound MatP in the chromosome replication termination region, *ter*, results in depletion of MukBEF from *ter*, a process essential for efficient daughter chromosome individualization and for preferential association of MukBEF with the replication origin region. Chromosome-associated MukBEF complexes also interact with topoisomerase IV (ParC$_2$E$_2$), so that their chromosome distribution mirrors that of MukBEF. We demonstrate that MatP and ParC have an overlapping binding interface on the MukB hinge, leading to their mutually exclusive binding, which occurs with the same dimer to dimer stoichiometry. Furthermore, we show that *matS* DNA competes with the MukB hinge for MatP binding. Cells expressing MukBEF complexes that are mutated at the ParC/MatP binding interface are impaired in ParC binding and have a mild defect in MukBEF function. These data highlight competitive binding as a means of globally regulating MukBEF-topoisomerase IV activity in space and time.

## Introduction

Structural Maintenance of Chromosomes (SMC) complexes play central roles in chromosome organization and segregation in all domains of life. The *Escherichia coli* SMC complex, MukBEF, is required for normal chromosome dynamics, and its absence leads to temperature-sensitive growth in rich medium, under conditions of overlapping replication cycles, and anucleate cell formation during permissive growth (*Hiraga et al., 1989*; *Nolivos and Sherratt, 2014*). Live-cell imaging of Muk⁻ strains growing under permissive conditions demonstrates disorganized chromosome locus positioning and failures to individualize newly replicated sister chromosomes as a consequence of impaired segregation to opposite cell halves (*Danilova et al., 2007*; *Badrinarayanan et al., 2012b*; *Badrinarayanan et al., 2012a*; *Mäkelä and Sherratt, 2020*). Wild-type (WT) cells contain ~100 functional MukBEF complexes, with 40–50% associated with chromosomes (*Badrinarayanan et al., 2012b*). These complexes are enriched at the replication origin (*ori*) region of the cell and act to position *ori*s at either the midcell in new-born cells or the cell quarter position in cells undergoing replication (*Danilova et al., 2007*; *Badrinarayanan et al., 2012b*; *Badrinarayanan et al., 2012a*; *Mäkelä and Sherratt, 2020*).

MukBEF complexes, which are largely restricted to *Escherichia coli* and its γ-proteobacterial relatives, have evolved several distinctive structural and functional features. MukBEF action in vivo, likely through extrusion of DNA loops, as demonstrated for a number of eukaryotic SMC complexes (*Davidson and Peters, 2021*), leads to the formation of a MukBEF axial core to the chromosome,

from which DNA loops of 15–50 kb emanate (*Mäkelä and Sherratt, 2020*). Although MukBEF architecture exhibits many of the conserved features present in prokaryote and eukaryote SMC complexes (*Figure 1A*), its uniquely dimeric kleisin, MukF, directs the formation of dimer of dimer complexes that form in vitro and in vivo (*Badrinarayanan et al., 2012b*; *Rajasekar et al., 2019*). Furthermore, the presence of MukBEF correlates with the presence of a number of other co-evolved genes and their proteins that include SeqA, Dam, AcpP, topoisomerase IV (topoIV) and MatP (*Brézellec et al., 2006*; *Mercier et al., 2008*; *Prince et al., 2021*; *Li et al., 2010*; *Hayama and Marians, 2010*). MatP interacts with 23 *matS* sites (13 bp) distributed throughout the 800 kb replication termination region (*ter*) of the chromosome (*Mercier et al., 2008*). MatP also interacts with the divisome component, ZapB, and MatP action through its binding to *matS* has been implicated in organizing *ter* and ensuring it is positioned correctly at the midcell prior to cell division (*Espéli et al., 2012*). Interaction in vivo of functional MukBEF complexes with *matS*-bound MatP leads to dissociation of MukBEF from *matS* sites, and the consequent displacement of MukBEF from the whole *ter* region, a process required for MukBEF to preferentially associate with *ori* and for efficient segregation of daughter chromosomes to opposite cell halves (*Mäkelä and Sherratt, 2020*; *Nolivos et al., 2016*; *Mäkelä et al., 2021*). An initial characterization in vitro showed that MatP binds to the isolated *E. coli* MukB dimerization hinge domain (*Nolivos et al., 2016*), although, the mechanistic details of how MatP-*matS* complexes lead to MukBEF displacement from *ter* remain to be determined. In the absence of MatP, MukBEF forms circular axial cores around the whole chromosome, leading to chromosome rotation and aberrations in the patterns of chromosome segregation (*Mäkelä and Sherratt, 2020*; *Mäkelä et al., 2021*).

The MukB hinge also interacts specifically with the ParC subunit of $ParC_2ParE_2$ topoIV heterotetramers, stimulating its catalytic activity and additionally directing it to sites of MukBEF action on the chromosome (*Li et al., 2010*; *Hayama and Marians, 2010*; *Hayama et al., 2013*; *Vos et al., 2013b*; *Zawadzki et al., 2015*; *Nicolas et al., 2014*). Consistent with this, at least 15 cellular topoIV molecules (~20% of the total) are associated with ~40 chromosome-bound MukBEF complexes at any time (~40% of the total), as measured by colocalization of MukBEF and topoIV fluorescent foci, and by single-molecule tracking in photoactivated localization microscopy (PALM) (*Badrinarayanan et al., 2012b*; *Zawadzki et al., 2015*). TopoIV is the major cellular decatenase and acts to unlink replicative catenanes as they form as replication progresses (*Adams et al., 1992*; *Peng and Marians, 1993*; *Zechiedrich et al., 1997*). Removal of topoIV activity in vivo leads to a complete failure to segregate newly replicated DNA, with consequent failure to divide normally, although replication continues unabated (*Wang et al., 2008*). TopoIV can also relax supercoiled DNA, with MukB-stimulated catalysis being directed to molecules with a right-handed chirality, replicative catenanes, and negative supercoils (*Li et al., 2010*; *Hayama and Marians, 2010*; *Hayama et al., 2013*; *Kato et al., 1992*). Previous work has shown that MukBEF complexes form active *ori*-associated clusters in the absence of topoIV, while in the absence of MukBEF, topoIV is functional, but cells have defects specifically in sister *ori* segregation, as measured by an increased cohesion time after replication, consistent with delayed decatenation (*Nolivos et al., 2016*; *Wang et al., 2008*). We propose that combined topoIV and MukBEF action may help coordinate chromosome unlinking with establishing the organization of newly replicated sister chromosomes.

In order to understand more about the functional interplay between MukBEF, topoIV, and MatP, we undertook an extensive characterization of the binding of MatP and topoIV to the MukB hinge in vitro. Using a range of independent ensemble and single-molecule assays, we showed that a dimeric MukB hinge binds a MatP dimer, a ParC dimer, or a single topoIV heterotetramer ($ParC_2E_2$) with comparable affinities. This raises the possibility of coordination of DNA passage between the 'gates' formed by both ParC and the MukB hinge. Importantly, we demonstrated that binding of MatP and ParC (or topoIV) to the MukB hinge is mutually exclusive, with them interacting with an overlapping interface on the hinge, thereby supporting an earlier inference of competitive binding in vivo (*Nolivos et al., 2016*; *Nicolas et al., 2014*). Cells expressing a MukB variant, which no longer interacts with topoIV and MatP in vitro were Muk+ as assessed by growth at 37 °C in rich medium, had no detectable cell division defect, and a modest defect in segregation of newly replicated *ori*s. This indicates that topoIV-dependent decatenation was proceeding efficiently in these cells. Quantitative live-cell imaging and single-molecule tracking showed that the variant hinge no longer interacts with ParC in vivo. Furthermore, mutants in MatP that fail to bind *matS* were impaired in hinge binding, consistent with the observation that interaction of MatP with the MukB hinge in vitro was inhibited

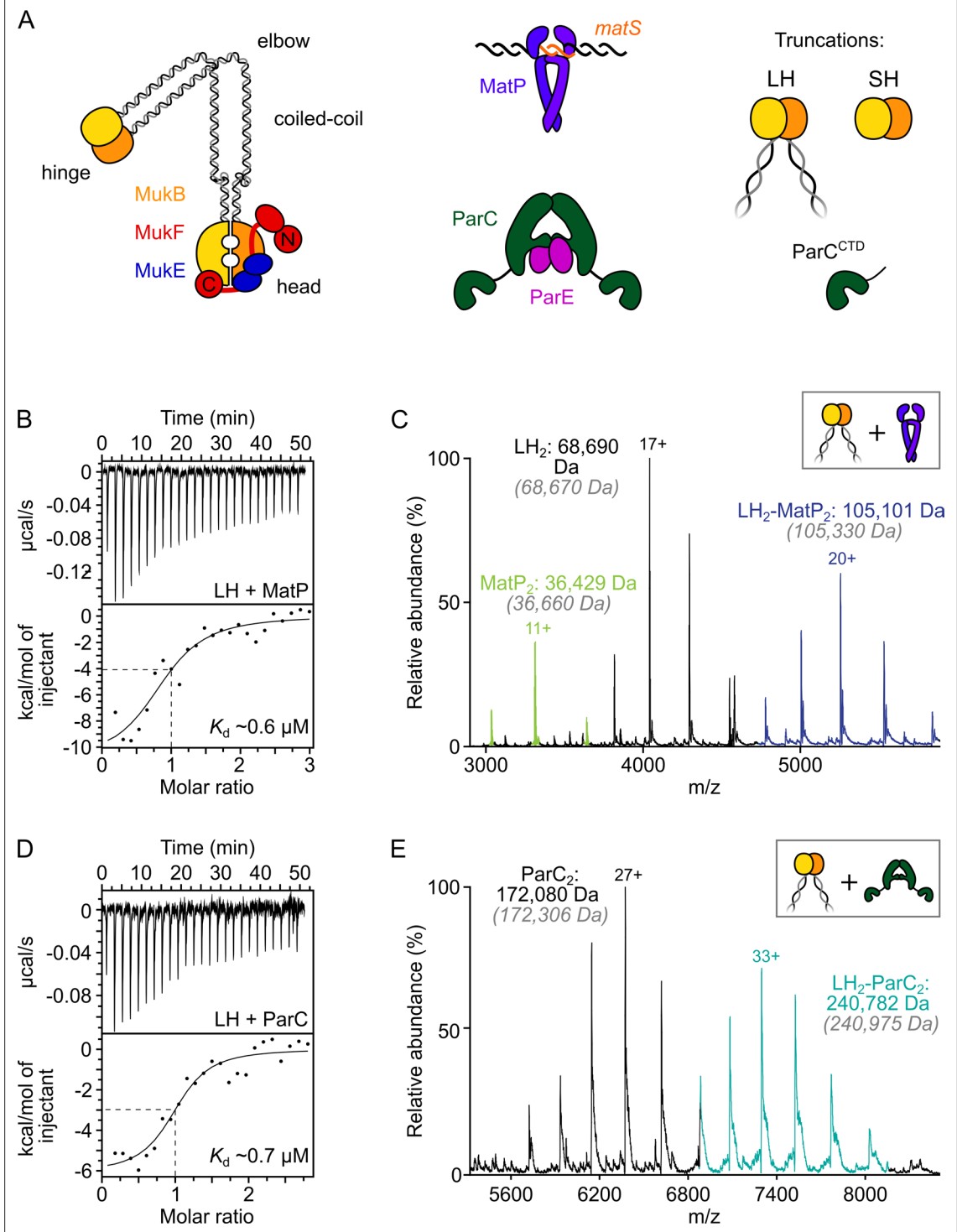

**Figure 1.** ParC dimers and MatP dimers each bind the dimeric MukB hinge with a 1:1 stoichiometry. (**A**) Schematics of the basic units of MukBEF complexes, topoIV and a MatP$_2$-*matS* complex. (**B**) Example isothermal titration calorimetry (ITC) raw thermogram and binding isotherm following titration of 100 µM MatP into 10 µM long hinge (LH) at 25 °C. The raw data from each experiment were fitted to a one-site model. n = 0.96 ± 0.08 (± SD), $K_d$ = 0.59 ± 0.04 µM (± SD), ΔH = –16.8 ± 2.2 (± SD) kcal/mol and ΔS = –27.8 ± 4.1 (± SD) cal/mol/deg. Three experimental repeats. (**C**) Native mass spectrometry (nMS). MatP was incubated at a twofold molar excess with LH before injection. An example spectrum is shown with the only detected LH-MatP complex highlighted in purple. Gray italics denote the theoretical mass of complexes. (**D**) ITC raw thermogram and binding isotherm following titration of 100 µM LH into 10 µM ParC at 25 °C. The raw data from each experiment were fitted to a one-site model. n = 0.84 ± 0.11 (± SD), $K_d$ = 0.69 ± 0.11(± SD) µM, ΔH = –7.48 ± 1.3 (± SD) kcal/mol and ΔS = –3.12 ± 1.4 (± SD) cal/mol/deg. Three experimental repeats. (**E**) nMS. ParC was incubated at a

*Figure 1 continued on next page*

*Figure 1 continued*

2-molar excess with LH before injection. An example spectrum is shown with only the $LH_2$-$ParC_2$ complex (1:1) detected and highlighted in cyan.

The online version of this article includes the following figure supplement(s) for figure 1:

**Figure supplement 1.** MatP dimers interact with the MukB hinge domain.

in the presence of *matS*-containing DNA. Since displacement of MukBEF complexes from *ter* in vivo requires that the MukBEF complexes interact, at least transiently, with MatP-*matS,* the failure of MatP-*matS* to bind the hinge is indicative of the hinge-MatP complex characterized here being a transient 'handover' intermediate. In parallel with this work, a separate MatP-*matS* binding site in the MukB coiled-coils was reported after the work here was completed (*Bürmann et al., 2021*), consistent with the proposal above.

## Results

### MatP dimers and ParC dimers bind the MukB dimeric hinge domain with the same 1:1 stoichiometry

We previously established that MatP dimers bind to the dimerization 'hinge' domain of MukB, although the precise details of this interaction were not characterized (*Nolivos et al., 2016*). Here we have used three independent assays to determine the stoichiometry and binding affinity of MatP to the MukB hinge. We exploited a MatP variant, MatPΔ18C, that carries a deletion of 18 C-terminal amino acid residues because it was more amenable for biochemical studies than the full-length protein; we refer to MatPΔ18C as 'MatP' hereafter. Like WT MatP, this protein is dimeric, binds *matS* sites and the MukB dimerization hinge, while a variant, MatPΔ20C, with two further C-terminal residues removed, retains the WT MatP ability to displace MukBEF complexes from *ter* in vivo (*Nolivos et al., 2016*).

Initial assays used a truncated MukB variant, 'long hinge' (LH; amino acid residues 568–863), a stable dimer that encompasses the dimeric globular hinge domain and 20% of the coiled-coil region (*Figure 1A*, *Figure 1—figure supplement 1A and B*). A 1:1 stoichiometry of binding of MatP to LH (one LH dimer binds one MatP dimer) ($K_d$ of 0.59 μM ± 0.04) was obtained in isothermal titration calorimetry (ITC) assays (*Figure 1B*). The same 1:1 stoichiometry was determined in native mass spectrometry (nMS) (*Figure 1C*) and by analytical ultracentrifugation (AUC) (*Figure 1—figure supplement 1C*). To test whether MukB hinge dimerization is necessary for MatP binding, we utilized the observation that a further hinge variant, 'short hinge,' lacking the entire coiled-coil (SH; amino acid residues 667–779) (*Figure 1—figure supplement 1A*), forms mixtures of monomers and dimers (*Figure 1—figure supplement 1D*). Complexes of MatP dimers with both SH monomers and dimers were observed in nMS (*Figure 1—figure supplement 1E*), demonstrating that dimerization of the MukB hinge is not essential for MatP dimer binding.

We then analyzed ParC binding to LH using ITC and nMS. In both assays, we measured a stoichiometry of 1:1 for ParC dimer-LH dimer complexes (*Figure 1D and E*). The affinity of the interaction, determined by ITC (0.69 ± 0.11 μM), was similar to that previously reported (*Li et al., 2010*) and to the affinity between MatP and LH measured here. This result is consistent with work, which showed binding of two monomeric ParC C-terminal domains to a dimeric MukB hinge truncation comparable to LH (*Li et al., 2010*). Nevertheless, our measured 1:1 stoichiometry for complexes of ParC dimers with MukB hinge dimers contrasts with that determined in a previous study, which reported that a single MukB dimer bound two dimeric ParC molecules (*Li et al., 2010*). We are confident that our determinations of the 1:1 stoichiometry, using two independent assays, are robust, and furthermore, nMS with a molar excess of ParC (1:8) still yielded a 1:1 stoichiometry (*Figure 1—figure supplement 1F*). We conclude that MatP dimers and ParC dimers each bind to the MukB dimeric hinge domain with comparable affinities, forming complexes of 1:1 stoichiometry.

### MatP and ParC/topoIV competitively interact with the MukB hinge

Earlier in vivo analyses led us to infer that topoIV and MatP might compete for binding to MukB. This was because fluorescent MukBEF complexes containing the mutated variant MukB[E1407Q] (hereafter MukB[EQ]), which binds ATP, but is hydrolysis impaired (*Hirano and Hirano, 2004*; *Hu et al., 2011*), were enriched at *matS* sites, dependent on the presence of MatP, but not detectably associated with topoIV

(*Badrinarayanan et al., 2012b*; *Nolivos et al., 2016*; *Nicolas et al., 2014*). In contrast, topoIV was associated with WT ATP-hydrolysis-competent MukBEF complexes associated with the chromosomal replication origin region (*ori*) (*Nicolas et al., 2014*).

To investigate whether MatP and ParC compete for binding to the MukB hinge in vitro, we first tested whether MatP, ParC, and LH could form ternary complexes using analytical size-exclusion chromatography (SEC). Initial experiments used the monomeric ParC C-terminal domain (ParC$^{CTD}$), which contains the MukB-binding interface (*Li et al., 2010*), so that molecular mass differences between complexes would be more readily resolved. When sampled in analytical SEC individually, LH dimers, MatP dimers, and ParC$^{CTD}$ all generated a single absorbance peak coincident with an elution volume of ~11.5 mL, ~15 mL, and ~14 mL, respectively (*Figure 2A*). LH, MatP, and ParC$^{CTD}$ were co-incubated at a 1:2:2 ratio at µM concentrations for 1 h prior to injection. The elution profile of the LH-MatP-ParC$^{CTD}$ sample indicated no formation of a larger species (typically interpreted by a shorter retention time); only the binary LH$_2$-MatP$_2$ and LH$_2$-ParC$^{CTD}$ complexes were present at approximately similar ratios (*Figure 2A*). To ascertain if any changes to the hydrodynamic radii of species influenced the sample elution profile and potentially obscured detection of a ternary complex of different shape, we also analyzed LH-MatP-ParC$^{CTD}$ mixtures by nMS. More importantly, nMS offers the means to detect more transient complexes not suitable for detection in analytical SEC. No ternary (LH$_2$-MatP$_2$-ParC$^{CTD}$) complexes were detected using nMS (*Figure 2B*). Consistent with these observations, when full-length ParC dimers were used instead of ParC$^{CTD}$, the binary MatP$_2$-LH$_2$ and ParC$_2$-LH$_2$ complexes were abundant, but only traces of possible LH$_2$-MatP$_2$-ParC$_2$ complexes were detected (*Figure 2C*). To test whether this competition was also evident for topoIV, we reconstituted ParC$_2$E$_2$ heterodimers and analyzed their ability to bind to LH dimers in the presence of equal amounts of MatP. ParC$_2$E$_2$ complexes interacted with LH dimers, but higher-order complexes that included MatP were absent (*Figure 2D*). We also detected a low abundance of complexes with the stoichiometry LH$_4$-ParC$_2$E$_2$; we propose these arise from higher order coiled-coil interactions, with their functional significance, if any, remaining unclear. Finally, the specific binding of *matS*-containing DNA to MatP (*Figure 2—figure supplement 1A*) did not allow for the formation of LH$_2$-MatP$_2$(-*matS*)-ParC$_2$ complexes (below) (*Figure 2—figure supplement 1B*). Complexes of non-specific DNA with MatP were not detected in nMS (*Figure 2—figure supplement 1A*).

To analyze the competition between ParC and MatP for binding to the MukB hinge more quantitatively, we exploited fluorescence correlation spectroscopy (FCS) using a Cy3B-labeled MatP variant engineered by the introduction of a cysteine residue ahead of the His-tag at the C-terminus (MatP has no intrinsic cysteines) (*Figure 3—figure supplement 1A*). As a control, we used MukB labeled internally at residue 718 with TAMRA conjugated to the unnatural amino acid p-azidophenylalanine. This was functional as assessed by ATPase assays of the labeled protein in vitro and by the in vivo phenotype of cells expressing a variant containing a S718F substitution (*Figure 3—figure supplement 1B and C*). MukB bound MatP-Cy3B with a similar affinity ($K_d$ ~0.25 µM) to that observed for LH binding to MatP (*Figure 3—figure supplement 1D*), indicating that this labeled MatP variant was not impaired and that the conformation of the hinge in LH is comparable to that of the hinge in full-length MukB. As expected, the interaction between MatP-Cy3B and MukB was competed out when a 50-fold molar excess of unlabeled MatP was added (*Figure 3A*). A threefold decrease in binding was observed when the reaction was challenged with a 10-fold excess of either ParC or ParC$^{CTD}$ (*Figure 3A*, with p-values = 3.12 × 10$^{-05}$ and 3.53 × 10$^{-03}$, respectively), while ParC with R705E and R729A substitutions (ParC$^{EA}$), a mutant defective in MukB hinge binding (*Hayama and Marians, 2010*; *Figure 3—figure supplement 1F*), did not significantly decrease MatP-Cy3B binding to MukB (p-value = 6.02 × 10$^{-2}$). No interactions between MatP and ParC were detected by FCS or nMS (*Figure 3—figure supplement 1G and H*).

Given the demonstration that ParC and MatP compete for binding to the MukB hinge, we explored whether MatP and ParC share the same or overlapping binding sites on the hinge by analyzing the interaction of a MukB variant known to be deficient in ParC binding, D697K D745K E753K (*Hayama et al., 2013*; LH$^{KKK}$ and MukB$^{KKK}$ hereafter), with MatP. We first assessed whether LH$^{KKK}$ and MatP could interact using native PAGE. Under the conditions used, LH, LH$^{KKK}$, and ParC$^{CTD}$ all formed well-defined singular species that were amenable to migration under electrophoresis (*Figure 3B*; note LH$^{KKK}$ migrated more slowly than wild-type LH, presumably because of the charge change). In contrast, MatP alone did not enter the Tris-based native gels, whereas its complexes with LH migrated as

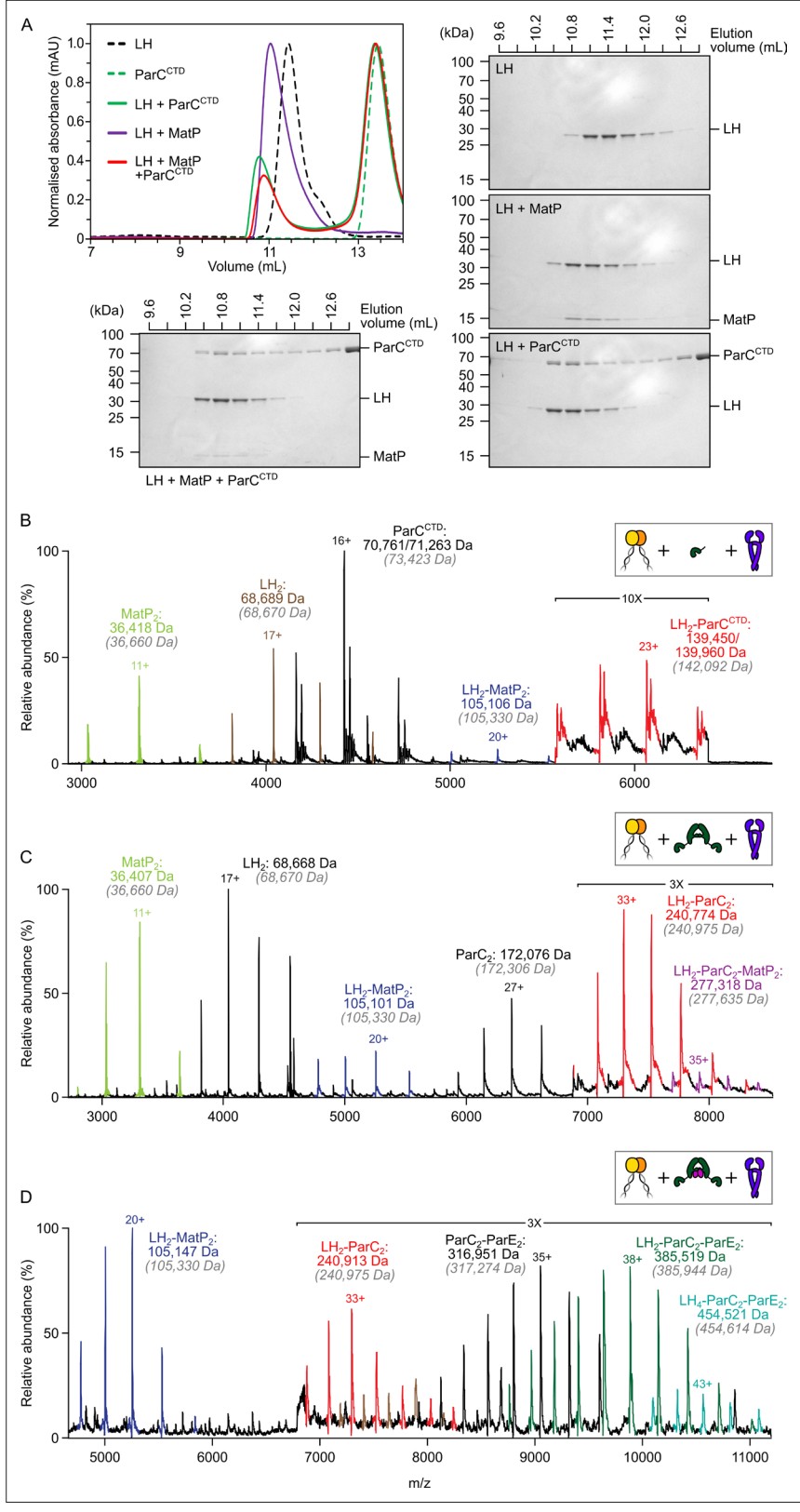

**Figure 2.** The MukB hinge does not form ternary complexes with MatP dimers and topoIV. (**A**) Analytical size-exclusion chromatography (SEC). Long hinge (LH), MatP, and ParC^CTD were co-incubated at a 1:2:2 ratio at µM concentrations for 1 h prior to injection and separated on a Superose 6 Increase column (left). 300 µL elution fractions were analyzed by SDS-PAGE and Coomassie staining (bottom and right). Note that ParC^CTD retains its

*Figure 2 continued on next page*

*Figure 2 continued*

MBP-N-terminal fusion (~46 kDa) for reasons of stability (see Materials and methods). (**B–D**) Representative mass spectra of complexes detected when co-incubating LH and MatP at a 1:2 ratio with two equivalents of either ParC$^{CTD}$ (**B**), ParC alone (**C**), or topoIV heterotetramers (**D**). Regions of the spectra are magnified, where indicated, to highlight detection of less abundant species. Gray italics denote the theoretical mass of complexes.

The online version of this article includes the following source data and figure supplement(s) for figure 2:

**Source data 1.** Raw data and uncropped gels for *Figure 2A*.

**Source data 2.** Raw data for analytical size-exclusion chromatography of *Figure 2A*.

**Figure supplement 1.** The MukB hinge does not form ternary complexes with MatP-*matS* complexes and topoIV.

a single band. ParC$^{CTD}$, similarly formed a resolvable complex with retarded migration when incubated with LH, while LH$^{KKK}$ failed to form a complex with ParC$^{CTD}$, as expected from previous work (*Hayama et al., 2013*). LH$^{KKK}$ was also impaired in MatP binding; a small reduction of free LH$^{KKK}$ was observed at a 1:2 ratio of LH$^{KKK}$:MatP, while at a 1:4 ratio, the reduction in free LH$^{KKK}$ was increased somewhat with a smear running more slowly, consistent with migration of unstable complexes. This interaction appeared to be abolished under conditions of analytical SEC (*Figure 3C*). The impaired binding of MatP to LH$^{KKK}$/MukB$^{KKK}$ was confirmed by titrating full-length MukB$^{KKK}$ against MatP-Cy3B in FCS (*Figure 3—figure supplement 1D*). In the latter case, the $K_d$ was increased from ~0.25 μM to ~1.6 μM. This interaction defect was not likely attributable to global misfolding as assayed by circular dichroism (CD) analysis of LH$^{KKK}$ (*Figure 3—figure supplement 1E*). We conclude that topoIV and MatP compete for binding to the MukB hinge as a consequence of MatP and ParC having overlapping binding sites on the MukB hinge.

## The MukB hinge fails to stably bind MatP-*matS* complexes

Since the action of MatP in displacing MukBEF complexes from *ter* in vivo requires that MatP is at least transiently bound to *matS* sites (*Mäkelä and Sherratt, 2020*; *Nolivos et al., 2016*), we initially anticipated that MatP-*matS* complexes would form stable complexes with the MukB dimerization hinge. We were therefore surprised to observe that addition of an excess of a 50 bp DNA fragment carrying an internal 13 bp *matS2* site (*Mercier et al., 2008*) almost totally abolished binding of MatP-Cy3B to MukB in FCS, while a non-specific DNA fragment of the same length and GC content had little or no effect on binding (*Figure 3A*). Furthermore, when we incubated MatP-*matS* complexes with LH, we failed to observe ternary LH-MatP-*matS* complexes in nMS, although the binary MatP-*matS* and LH-MatP complexes were present (*Figure 4A*). Using nMS, we confirmed that, under these conditions, MatP was specifically bound to *matS* (*Figure 2—figure supplement 1A*). Similarly, in native PAGE, we identified the binary, but not the ternary complexes (*Figure 4B*). LH and MatP formed a discrete complex, which was disrupted in the presence of *matS*, while LH ran with the expected migration for the protein alone, and MatP-*matS* formed a fast migrating nucleoprotein complex containing *matS* DNA. This change in mobility likely reflects both the conformational changes MatP undergoes upon *matS* binding (*Dupaigne et al., 2012*) and the negative charge of the bound DNA. Free 50 bp *matS* DNA migrated off the gel. The presence of the 50 bp non-specific DNA fragment that did not stably bind MatP in nMS (*Figure 2—figure supplement 1A*) partially impaired MatP binding to LH (*Figure 4B*). Some non-specific DNA binding to MatP still occurred under these conditions and led to smeared complexes containing both MatP and DNA (*Figure 4—figure supplement 1A*). No stable complexes of LH with 50 bp DNA were detected in nMS (*Figure 4—figure supplement 1B*).

Since small-angle X-ray scattering (SAXS) analysis of the MatP envelope indicated that MatP alone is less compact than MatP-*matS*, whose structure has been determined by X-ray crystallography (*Dupaigne et al., 2012*), we considered whether these global conformational differences are responsible for the observed differential binding of MatP and MatP-*matS* to the MukB hinge. However, given that MatP binding to *matS* and LH is mutually exclusive, we explored experimentally the alternative hypothesis that *matS* and the MukB hinge share an overlapping binding interface on MatP. The structure of MatP-*matS* identified residues involved in DNA binding, including K71, Q72, R75, and R77 (*Dupaigne et al., 2012*). Consistent with this, a quadruple substitution of these residues with Ala (MatP$^{4A}$), which neutralized their charge, or with Glu (MatP$^{4E}$), which reversed it, led to the impairment or complete loss of specific binding with *matS* DNA, respectively (*Figure 4—figure supplement*

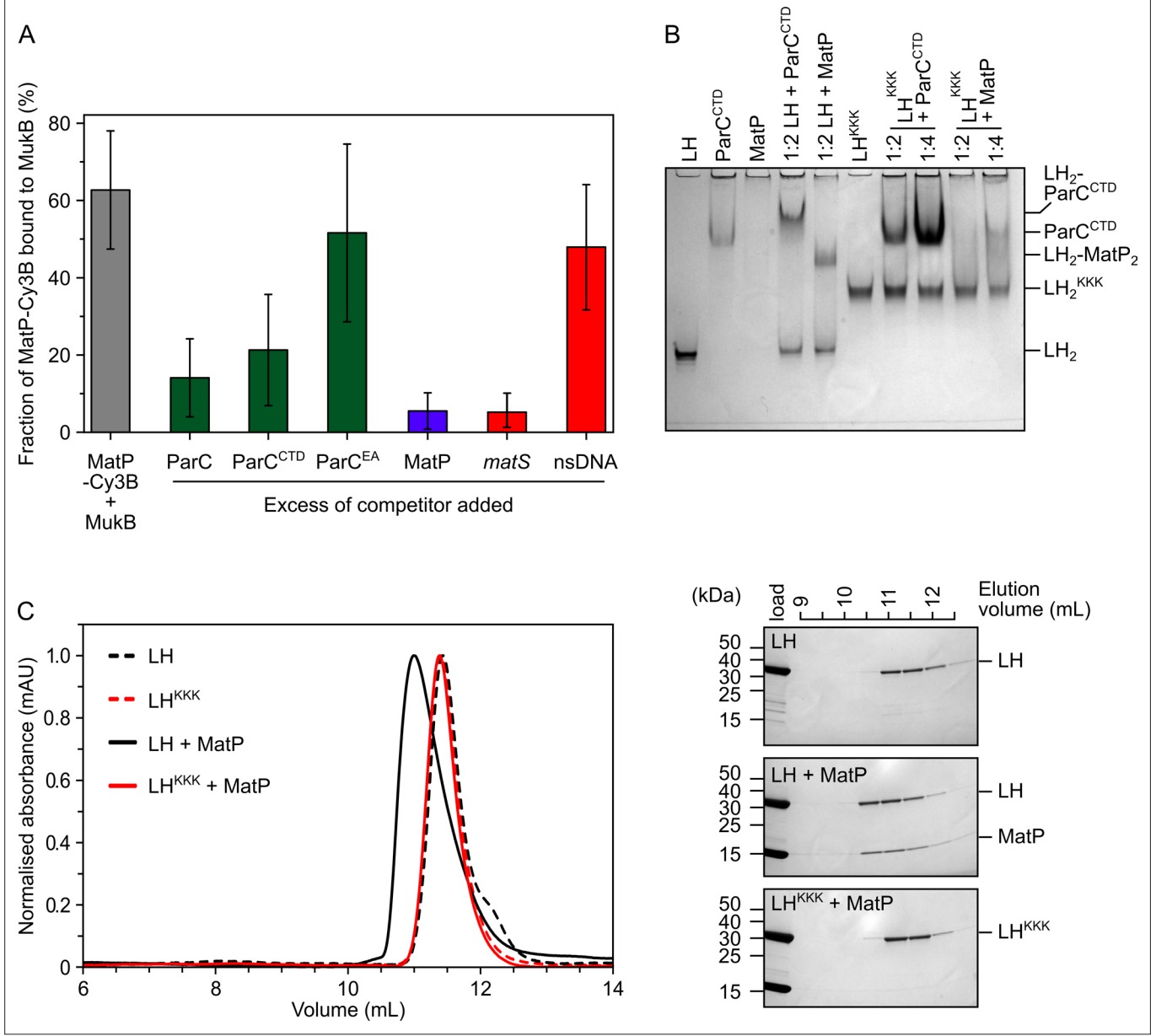

**Figure 3.** MatP and ParC compete for binding to overlapping sites on the MukB hinge. (**A**) Fluorescence correlation spectroscopy (FCS) measurements of competition for binding between ParC and MatP for MukB and also between MatP and 15 bp DNA hairpins, containing a 13 bp *matS2* site or non-specific sequence, for MukB. Cy3B-labeled (MatP) was fixed at 1 nM and wild-type (MukB) at 400 nM; this achieved ~60% binding of all MatP-Cy3B to MukB. Autocorrelation curves were fit to a two-component equation (Equation 2) with the diffusion time of Cy3B-MatP fixed to 4.5 ms ($\tau_1$), whilst the diffusion time ($\tau_2$) of bound complex was allowed to float to obtain the best fitting for the data. All ParC variants, unlabeled MatP, and DNA were added at a 10-, 50-, and 2.5-fold molar excess, respectively. Error bars represent mean ± SD. p-values were determined from two-tailed two-sample t-tests to assess the significance of the effect of competitors upon binding of MatP to MukB: $3.12 \times 10^{-05}$, $3.53 \times 10^{-03}$, and $6.02 \times 10^{-02}$, for ParC, ParC$^{CTD}$, and ParC$^{EA}$, respectively. (**B**) Native PAGE. Varying ratios of MatP or ParC$^{CTD}$ were incubated with long hinge (LH) (at 3 µM) for 30 min on ice prior to electrophoresis under non-denaturing conditions. All proteins were run alone as a reference for their mobility in an 8% tris-glycine gel. Note, MatP alone poorly enters the gel. Stoichiometries of the complexes formed are inferred from corresponding native mass spectrometry (nMS) data (***Figure 1C and E***, ***Figure 2B***). (**C**) Analytical size-exclusion chromatography (SEC). Either wild-type LH or the LH$^{KKK}$ variant, which is defective in ParC binding (***Hayama et al., 2013***), was co-incubated with MatP at a 1:2 ratio with LH (at 20 µM) for 1 h on ice prior to injection and separated on a Superose 6 Increase column (left). 500 µL elution fractions were analyzed by SDS-PAGE and Coomassie staining (right). Free MatP dimers elute as a peak at ~15 mL.

The online version of this article includes the following source data and figure supplement(s) for figure 3:

**Source data 1.** Raw data and uncropped gel for ***Figure 3B***.

**Source data 2.** Raw data and uncropped gels for ***Figure 3C***.

*Figure 3 continued on next page*

*Figure 3 continued*

**Source data 3.** Raw data for analytical size-exclusion chromatography of *Figure 3C*.

**Figure supplement 1.** MatP and ParC compete for binding to overlapping sites on the MukB hinge.

**Figure supplement 1—source data 1.** Raw and fitted data for autocorrelation curves in *Figure 3—figure supplement 1A*.

**Figure supplement 1—source data 2.** Raw data for triplicate ATPase assay in *Figure 3—figure supplement 1B*.

**Figure supplement 1—source data 3.** Tabulated data including fitted data for *Figure 3—figure supplement 1D*.

**Figure supplement 1—source data 4.** Raw data of that presented in *Figure 3—figure supplement 1E*.

*1A*). MatP[4A] retained a partial ability to interact with the non-specific DNA fragment, while MatP[4E] showed no detectable binding to non-specific DNA (*Figure 4—figure supplement 1A*). MatP[4A] was partially defective in LH binding, as judged by the limited ability of it to shift free LH to a more slowly migrating complex. Again, MatP[4E] was fully impaired in its interaction with LH, consistent with MatP using overlapping determinants to bind the MukB hinge and *matS* (*Figure 4B*). To test whether MatP[4E] or MatP[4A] have altered overall structures that prevent hinge binding, we showed that both MatP[4A] and MatP[4E] are dimeric and generate a Gaussian distribution of charge states in nMS, indicative of folding (*Figure 4—figure supplement 1C*). Therefore, we favor a model in which *matS* and the hinge share a common binding interface on MatP.

## Cells expressing a MukB hinge mutant defective in ParC and MatP binding in vitro are impaired in MukB-ParC interaction in vivo and have impaired MukBEF function

Given that the MukB[KKK] hinge mutant is deficient in both ParC and MatP binding in vitro, we anticipated that cells expressing MukB[KKK] might exhibit both a chromosome segregation-cell division phenotype, resulting from the lack of correct targeting and catalysis by ParC/topoIV, and a MatP- phenotype. Moreover, we considered that this mutant might additionally have a MukB- temperature-sensitive growth phenotype since temperature-sensitive growth was reported for a different MukB hinge mutant (MukB[D692A]) that was also impaired in ParC binding in vitro (*Li et al., 2010*).

MukB[KKK]-expressing cells were not temperature-sensitive for growth in rich medium, indicative of no complete loss of MukBEF function (*Figure 5A*). Additionally, in epifluorescence imaging, Δ*mukB* cells expressing basal levels of MukB[KKK] from the multicopy plasmid pBAD24, with fluorescently labeled *ori1* and *ter3* loci and a functional chromosomal *mukE-mYPet* gene (*Mäkelä and Sherratt, 2020*; *Nolivos et al., 2016*), had almost identical median cell lengths to those expressing WT MukB (*Figure 5—figure supplement 1A*). In contrast, cells with major defects in decatenation/chromosome segregation resulting from the impaired action of topoIV are filamentous with nonsegregated chromosomes (*Wang et al., 2008*; *Grainge et al., 2007*). Therefore, we conclude that MukB[KKK]-expressing cells had no substantial defect in decatenation by topoIV. Furthermore, MukB[KKK]-expressing cells had an *ori1* locus number distribution intermediate between that of WT and Δ*mukB* cells, with the latter having a characterized delay in segregation of newly replicated *ori1* loci and therefore an increased proportion of cells containing a single *ori1* focus (*Nolivos et al., 2016*; *Wang et al., 2008*; *Figure 5—figure supplement 1B*). We have proposed that this is likely a consequence of impaired decatenation because modest topoIV overexpression reverses the delay in *ori1* segregation (*Wang et al., 2008*). Δ*matP* cells had an *ori1* locus distribution similar to that of Δ*mukB* cells, indicating that they too may have a defect in decatenation of newly replicated *ori1* loci, probably as a consequence of reduced association of MukBEF and topoIV with *ori1* in these cells.

To examine whether topoIV associates with MukB[KKK]EF complexes in vivo, we used single-molecule tracking in PALM using a functional PAmCherry-ParC fusion at the endogenous ParC locus since we have previously reported that interaction in vivo of topoIV molecules with immobile chromosome-associated MukBEF complexes leads to a higher proportion of topoIV molecules becoming immobile (*Zawadzki et al., 2015*). Cells expressing MukB[KKK]EF showed an almost identical distribution of immobile/mobile ParC molecules to cells lacking MukBEF, while control cells expressing WT MukB showed enrichment of MukBEF-bound topoIV molecules, as previously reported (*Zawadzki et al., 2015*; *Figure 5C*), thereby demonstrating that the topoIV-MukB interaction is ablated in vivo in the MukB[KKK] mutant. Furthermore, the same level of ParC molecule immobilization in MukB[KKK] and Δ*mukB* cells indicates that there are no further binding sites for ParC within the MukBEF complex. Taken

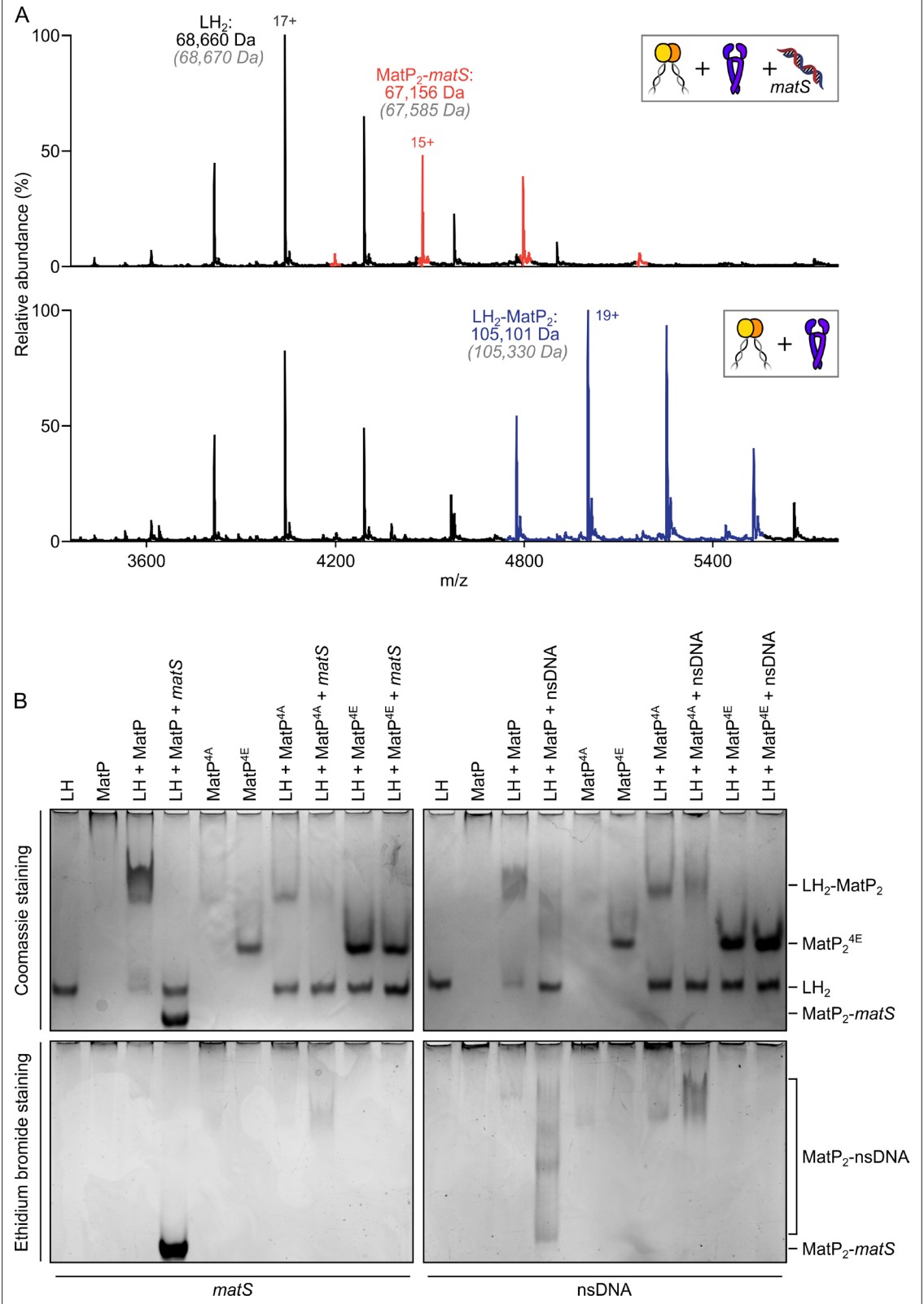

**Figure 4.** *matS* sites compete with the hinge for MatP binding. (**A**) Representative mass spectra of species detected between long hinge (LH) and MatP in the presence of *matS* DNA. LH, MatP, and in the case of the upper panel *matS*, were mixed at 1:2:1. Gray italics denote the theoretical mass of complexes. (**B**) Native PAGE. Formation of LH$_2$-MatP$_2$-*matS* ternary complexes were not detected. MatP$^{4A}$ (K71A, Q72A, R75A, and R77A) and MatP$^{4E}$ (K71E, Q72E, R75E, and R77E) are impaired in binding to the MukB hinge domain and to DNA. Samples were loaded onto two equivalent native gels

*Figure 4 continued on next page*

*Figure 4 continued*

– one for Coomassie staining and one for ethidium bromide staining. Stoichiometries of the complexes formed are inferred from corresponding native mass spectrometry (nMS) data (**A**, *Figure 4—figure supplement 1*, and *Figure 2—figure supplement 1A*).

The online version of this article includes the following source data and figure supplement(s) for figure 4:

**Source data 1.** Raw data and uncropped gels for *Figure 4B*.

**Figure supplement 1.** *matS* sites compete with the MukB hinge for MatP binding.

**Figure supplement 1—source data 1.** Raw data and uncropped gels for *Figure 4—figure supplement 1A*.

together, the normal median cell sizes, lack of discernible defects in cell division/nucleoid segregation, and modest effects on *ori1* segregation of cells impaired in the interaction between topoIV and MukBEF complexes indicate that any delays in chromosome unlinking by decatenation in these cells do not seriously impact viability, chromosome segregation, and consequent cell division.

To further ascertain the properties of MukB$^{KKK}$EF cells, we undertook a more detailed analysis of MukBEF behavior in relation to *ori1* and *ter3* locus positioning in the cell populations shown in *Figure 5B*. Control cells expressing basal levels of WT MukB from pBAD24 formed *ori1*-associated fluorescent MukBEF foci (*Figure 5B*), as reported previously (*Danilova et al., 2007*; *Badrinarayanan et al., 2012b*; *Mäkelä and Sherratt, 2020*; *Mäkelä et al., 2021*). Negative control Δ*mukB* cells containing just the pBAD24 vector showed no evidence of nucleoid-localized fluorescent MukBEF complexes (*Figure 5B*), consistent with their temperature-sensitive growth (*Figure 5A*). MukB$^{KKK}$EF-expressing cells formed somewhat diffuse fluorescent MukBEF complexes that were distinct in morphology from MukBEF complexes in WT and Δ*matP* cells, both of which form discrete well-defined foci (*Figure 5B*). We then compared the distribution of background-subtracted maximum fluorescent MukE pixel intensities in WT, *mukB$^{KKK}$*, Δ*matP*, and Δ*mukB* cells (*Figure 5—figure supplement 1C*). The brightest pixel of MukB$^{KKK}$EF complexes was intermediate in intensity between that of WT (p-value < $10^{-5}$) and Δ*mukB* cells (p-value $7 \times 10^{-4}$), consistent with the observation that MukB$^{KKK}$-expressing cells produce somewhat diffuse foci that are likely to be chromosome-associated; uniformly distributed fluorescence is characteristic of nonfunctional MukBEF complexes that are unable to stably associate with DNA (*Badrinarayanan et al., 2012b*; *Mäkelä and Sherratt, 2020*; *Nolivos et al., 2016*). Control Δ*matP* cells had a broad distribution of maximum pixel intensities that were significantly higher than that of the MukB$^{KKK}$EF-expressing cells (p-value 0.0034). Mean MukE pixel intensity was not significantly different between strains, consistent with the MukE-mYPet copy number being comparable in the strains tested (*Figure 5—figure supplement 1C*).

We then analyzed the distance of the brightest MukE pixel of MukB$^{KKK}$EF complexes to either the *ori1* or *ter3* locus in each cell and compared this distribution to that of the brightest pixel in control WT, Δ*mukB*, and Δ*matP* cells (*Figure 5D*, *Figure 5—figure supplement 1D*). We analyzed brightest pixel position rather that focus centroid because of the diffuse nature of MukB$^{KKK}$EF foci and the lack of foci in Δ*mukB* cells. Cells expressing WT MukBEF exhibit a preferential association of MukBEF complexes with the replication origin region (marked by *ori1*), arising directly from MukBEF depletion from *ter* as a consequence of the ATP hydrolysis-dependent MukBEF dissociation from the 23 MatP-bound *matS* sites within *ter* (*Nolivos et al., 2016*; *Figure 5D*). MukBEF complexes in Δ*matP* cells lose their preferential *ori1* association because of the failure to deplete MukBEF complexes from *ter3*. Consequently, MukBEF complexes in such cells are equally likely to be *ori1* and *ter3* associated (and with other loci tested) (*Mäkelä and Sherratt, 2020*; *Nolivos et al., 2016*; *Figure 5D*). In Δ*mukB* cells, the mean measured distances of the brightest MukE pixel to *ori1* and *ter3* were almost identical, as expected, since the brightest pixel is expected to be placed randomly in the cell, given the uniformly distributed fluorescence. MukB$^{KKK}$EF complexes had a reduced *ori1* association and increased *ter3* localization, with their pattern of distribution being more similar to Δ*mukB* cells than Δ*matP* cells (*Figure 5D*, *Figure 5—figure supplement 1D*), suggesting a partial defect in MukB function, despite some chromosomal association remaining. Since fluorescent *ori1* foci become mislocalized in Δ*mukB* cells, becoming positioned towards the older poles (*Danilova et al., 2007*; *Mäkelä et al., 2021*), we then compared *ori1* (and *ter3*) positioning in MukB$^{KKK}$ cells with that in WT, Δ*mukB*, and Δ*matP* cells. In WT cells containing two segregated *ori1* loci, the loci were positioned close to the cell quarter positions as reported previously (*Figure 5E*; *Danilova et al., 2007*; *Badrinarayanan et al., 2012b*; *Badrinarayanan et al., 2012a*; *Mäkelä and Sherratt, 2020*). MukB$^{KKK}$EF-expressing cells had their

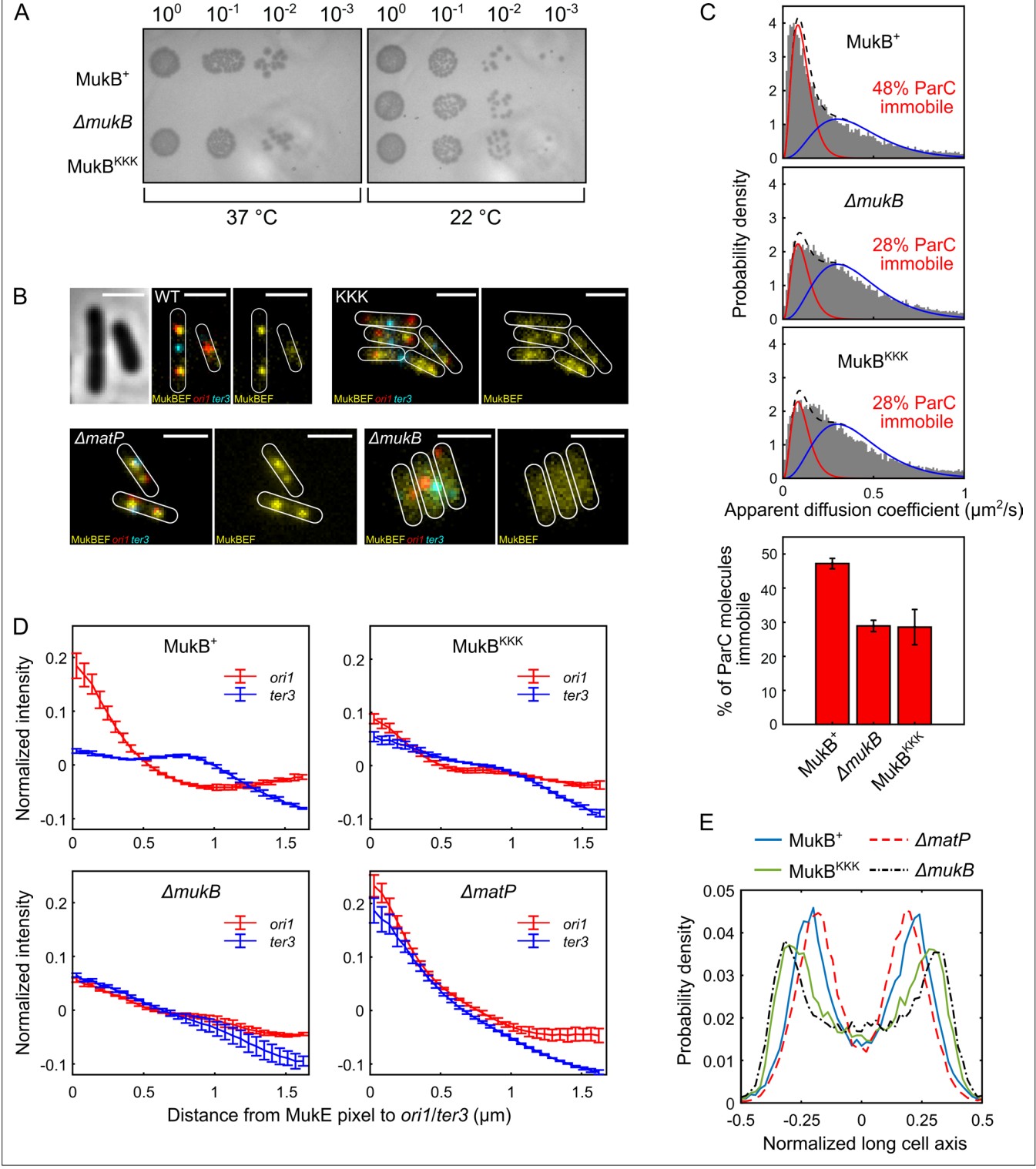

**Figure 5.** Cells expressing MukB[KKK] are impaired in ParC binding and exhibit defects in MukBEF function. (**A**) MukBEF phenotype of MukB[+], MukB[KKK], and Δ*mukB* cells as judged by temperature-sensitive growth in rich medium (LB) at 22 °C and 37 °C. Basal levels of plasmid-borne MukB and MukB[KKK] were expressed from cells carrying a chromosomal MukB deletion *Zawadzka et al., 2018*. Δ*mukB* control cells carried the empty plasmid. Two biological repeats gave the same result. (**B**) Representative fluorescence images with cell borders of Δ*mukB* cells with fluorescently labeled MukE (mYPet), *ori1*(mCherry), and *ter3* (mCerulean) (AU2118; *lacO240* @*ori1* (3908) (*hyg*), *tetO240@ter3* (1644) (*gen*), Δ*leuB::Plac-lacI-mCherry-frt*, Δ*galK::Plac-tetR-mCerulean-frt*, *mukE*-mYPet Δ*araBAD::FRT* (AraC+) FRT-T1-T2-Para-Δ*mukB*::kan) (*Mäkelä and Sherratt, 2020*; *Nolivos et al., 2016*), expressing basal levels of pBAD24 plasmid-borne WT MukB, MukB[KKK], and empty pBAD24 plasmid control (Δ*mukB*). Δ*matP* cells expressing MukBEF under the

*Figure 5 continued on next page*

*Figure 5 continued*

native *smtA-mukBEF* promoter, with fluorescently labeled MukE, *ori1*, and *ter3* labeled MukB, *ori1*, and *ter3* (SN302) (**Mäkelä and Sherratt, 2020**) Scale bars: 2 µm. Wild-type (WT) MukB, 4837 cells; MukB$^{KKK}$, 5846 cells; Δ*mukB*, 10,670 cells; Δ*matP*, 17,933 cells. For median cell lengths, see *Figure 5—figure supplement 1A*. (**C**) Single-molecule tracking (photoactivated localization microscopy [PALM]) of ParC-PAmCherry molecules in Δ*mukB* cells (PZ129) **Prince et al., 2021**; **Zawadzki et al., 2015** complemented with basal levels of plasmid-expressed WT MukB or MukB$^{KKK}$; control Δ*mukB* cells contained an empty plasmid. For each condition, the distribution of ParC apparent diffusion coefficients was fitted to a two-species model as in **Prince et al., 2021**; **Zawadzki et al., 2015**. Bar chart shows same data, with SD from three experimental repeats. (**D**) Normalized MukE pixel intensity as a function of distance to *ori1/ter3* in asynchronous populations in the strains in (**B**). WT MukB, 15,391 cells; MukB$^{KKK}$, 14,777 cells; Δ*mukB*, 22,807 cells; Δ*matP*, 22,623 cells. Error bars denote SD from three repeats. (**E**) Localization of *ori1* along normalized long cell axis in two *ori1* cells in the strains in (**B**). WT MukB, 7750 cells; MukB$^{KKK}$, 6417 cells; Δ*mukB*, 19,899 cells; Δ*matP,* 11,920 cells. Data from three biological repeats.

The online version of this article includes the following figure supplement(s) for figure 5:

**Figure supplement 1.** MukBEF distances to *ori1/ter3* and marker localizations.

*ori1* loci positioned towards the outer poles, similar to those of Δ*mukB* cells, a pattern distinct from that of Δ*matP* cells. Localization of *ter3* in these same two *ori1* focus cell populations showed the MukB$^{KKK}$ cells to have a similar pattern of *ter3* localizations to that of Δ*mukB* cells, with a strong preference for midcell positioning (*Figure 5—figure supplement 1E*). In cells with a single *ori1* focus, the same overall trend was seen, although less marked; MukB$^{KKK}$EF cells had localization patterns for *ori1* and *ter3* loci more similar to that of Δ*mukB* cells than Δ*matP* cells (*Figure 5—figure supplement 1F* and G, respectively).

Finally, we assessed the cellular position of the brightest MukE pixel in MukB$^{KKK}$EF-expressing cells and control WT, Δ*mukB,* and Δ*matP* cells (*Figure 5—figure supplement 1H*). WT and Δ*matP* cells have their MukBEF complexes positioned close to midcell (cells containing a single *ori1* focus) or in separate cell halves (cells with two *ori1* foci), as reported previously (**Badrinarayanan et al., 2012b**; **Mäkelä and Sherratt, 2020**; **Nolivos et al., 2016**). The pattern of brightest pixel localization in MukB$^{KKK}$EF-expressing cells was very similar to that of Δ*mukB* cells, indicating that the proposed ATP hydrolysis-dependent patterning system that positions MukBEF complexes on the chromosome at preferential cellular positions (**Murray and Sourjik, 2017**; **Hofmann et al., 2019**) is defective in *mukB*$^{KKK}$ cells, Taken together, the data lead us to conclude that MukB$^{KKK}$EF-expressing cells have a substantial defect in an uncharacterized MukBEF function. This defect could be in MukB ATP binding hydrolysis, DNA binding, or one of the uncharacterized conformations that must occur during the cycles of MukBEF action. Nevertheless, MukB$^{KKK}$EF complexes are sufficiently functional to form some chromosome-associated clusters and to render cells temperature-resistant in growth. Despite MukB$^{KKK}$EF complexes not interacting with topoIV in vivo, MukB$^{KKK}$EF expressing cells did not exhibit a severe defect in decatenation/chromosome segregation. The analyses did not reveal an in vivo defect in interaction of MukB$^{KKK}$EF complexes with MatP-*matS,* consistent with a second binding site for MatP-*matS* in the MukB coiled-coils (see Discussion).

## Discussion

The results presented here provide new insight into the functional interplay between MukBEF, MatP, and topoIV in the organization and processing of *E. coli* chromosomes, and potentially those of other γ-proteobacteria that have co-evolved the same orthologs. Many of these bacteria undergo overlapping replication cycles, have no identified chromosome segregation systems other than MukBEF, and have no characterized system that anchors chromosomes to the membrane for most of the cell cycle. In contrast, bacteria encoding canonical SMC complexes use ParAB-*parS* systems to facilitate efficient chromosome segregation, most frequently initiate and complete replication in a single cell cycle, utilize chromosome tethering to a pole, and organize their chromosomes so that the left and right replichores are arranged about the longitudinal axis of the cell (**Wang and Rudner, 2014**; **Reyes-Lamothe and Sherratt, 2019**). In *E. coli*, MukBEF complexes are positioned autonomously at either midcell in newborn cells or at the cell quarter positions in the remainder of the cell cycle by a Turing patterning system (**Murray and Sourjik, 2017**; **Hofmann et al., 2019**). The positioned MukBEF complexes associate with and position replication origins as a direct consequence of the MatP-*matS*-promoted depletion of MukBEF from *ter*, thereby ensuring the 'correct' placement of newly replicated chromosomes and their genetic loci within cells (**Mäkelä and Sherratt, 2020**). This

placement has left and right replichores on either side of *ori* in newborn cells and generates a left-*ori*-right-left-*ori*-right translational symmetry about the transverse cell axis after replication in most cells (*Mäkelä et al., 2021*; *Wang et al., 2005*; *Wang et al., 2006*; *Nielsen et al., 2006*). Intriguingly, *Vibrio cholerae* encodes MukBEF and MatP, but also uses separate ParAB-*parS* systems to segregate its two chromosomes (*Val et al., 2014*). Globally, bacterial topoIV otholgs remove replicative catenanes as they are generated during replication progression, while potentially helping to maintain supercoiling homeostasis (*Reyes-Lamothe and Sherratt, 2019*; *Postow et al., 2001*).

## TopoIV interaction with the MukB hinge

Previous work has shown that ParC dimers and topoIV $ParC_2E_2$ heterotetramers bind to the MukB dimerization hinge through one of the five ParC C-terminal 'blades,' the interaction leading to stimulation of topoIV catalysis (*Li et al., 2010*; *Hayama et al., 2013*; *Vos et al., 2013b*). Interaction of this same C-terminal blade with transfer segment DNA may direct the capture of a specific DNA topology during strand transfer, with MukB hinge binding leading to impairment of the interaction between $ParC^{CTD}$ and DNA in vitro (*Vos et al., 2013b*). Here we have demonstrated, using two independent assays, that only one intact ParC dimer, or one topoIV heterotetramer, binds a single dimeric hinge, whereas earlier work indicated that two ParC dimers bind one hinge dimer (*Li et al., 2010*). To explain the earlier result, it was proposed that a steric constraint prevents the two C-terminal domains of ParC, ~190 Å apart in the ParC crystal structure, from docking on the two binding sites separated by ~45 Å on a dimeric hinge, thereby leading to two ParC dimers bound to the hinge, with each having only one of its two CTDs bound (*Vos et al., 2013b*). Nevertheless, even with an eightfold molar excess of ParC in our nMS assay, we still observed only 1:1 ParC dimer:MukB hinge dimer complexes, with no evidence of a 2:1 stoichiometry, strengthening the conclusion that 1:1 is the physiologically relevant stoichiometry. The most logical explanation of this is that a single ParC dimer, either alone, or in a topoIV heterotetramer, has both of its C-terminal domains interacting with the two binding interfaces on a single dimeric MukB hinge. An alternative explanation, in which binding of one $ParC^{CTD}$ of a ParC dimer to one side of the hinge is incompatible with binding of a second $ParC^{CTD}$ to the second hinge interface, because of negative cooperativity or steric constraints, seems unlikely given that two isolated ParC CTDs bind independently to the two hinge binding interfaces (*Li et al., 2010*; *Vos et al., 2013b*). Our favored explanation requires that the ParC C-terminal domains adopt an alternative conformation in which they are closer together (feasible in light of the flexible linker between the CTD and N-terminal domain, *Figure 1A*) and/or the binding interfaces on the MukB hinge move apart by breaking or reorganizing the dimerization interface. Precedents for alternative conformations of the ParC C-terminal domains (and their equivalent in other topoisomerases) relative to the core enzyme have come from structural and modeling analyses of other bacterial type II topoisomerases (*Laponogov et al., 2013*; *Costenaro et al., 2005*; *Baker et al., 2011*; *Corbett et al., 2004*; *Corbett et al., 2005*). Furthermore, MukB hinge opening on association with a ParC dimer would be consistent with previous proposals for other SMC complexes that their hinge opening allows DNA passage into or out of the SMC ring (*Robison et al., 2018*; *Srinivasan et al., 2018*; *Gruber et al., 2006*). Assuming our proposal is correct, it leads to a scenario in which a topoIV heterotetramer bound to a MukB hinge will have one of its DNA passage gates sitting positioned above the hinge, whose own gate, formed by dimerization, potentially opening to allow DNA passage between topoIV and MukBEF. It is attractive to think that in these two 'multi-gate' protein complexes, regulated gate opening and closing may be used for coordinating topoIV and MukBEF action; for example, with newly decatenated sister DNA being transferred from topoIV into the MukB SMC ring, in order that each newly replicated-decatenated sister chromosome can be appropriately organized by MukBEF.

## Mutually exclusive binding of MatP and topoIV to the MukB dimerization hinge domain

Our demonstration that binding of MatP dimers and ParC dimers/topoIV heterotetramers to the dimeric MukB hinge is mutually exclusive provides an attractive potential means for the spatiotemporal regulation of topoIV activity by MukBEF and MatP. Consistent with this competition is our observation that mutations in the MukB hinge that ablate ParC binding also severely impair MatP binding to the hinge in vitro. Consistent with the competitive binding, previous reports showed that topoIV is not associated with $MukB^{EQ}$ complexes enriched at MatP-*matS* sites within *ter* as a consequence

of the ATP hydrolysis defect of MukB$^{EQ}$, but is associated with MukBEF complexes within *ter* in MatP⁻ cells (*Nicolas et al., 2014*). In contrast, colocalization of MatP and MukBEF was not observed; rather fluorescent MatP foci were associated with *ter* throughout the cell cycle through its binding to *matS* sites (*Mercier et al., 2008*). Therefore, it seems likely that whereas topoIV is frequently associated with chromosome-bound MukBEF complexes, MatP is at most transiently bound to such complexes, with much or most of it bound to *matS* sites.

## The MukB hinge and *matS* DNA compete for binding to MatP

We were initially surprised to observe that the MukB hinge and *matS* DNA compete for binding to MatP in vitro. This is because it has been previously established that MukBEF complexes interact with MatP bound to *matS* sites in vivo, leading to ATP hydrolysis-dependent MukBEF dissociation from *matS* sites (*Mäkelä and Sherratt, 2020*; *Nolivos et al., 2016*). We have proposed that this dissociation from up to 23 13 -bp MatP-bound *matS* sites (368 bp total) occurs during active DNA transport (probably by loop extrusion) by MukBEF, thereby leading to consequent MukBEF depletion from the whole *ter* region (*Mäkelä and Sherratt, 2020*). Nevertheless, our demonstration in vitro of mutually exclusive binding of *matS* DNA and an isolated MukB hinge to MatP is consistent with the demonstration that a MatP mutant defective in interaction with the MukB hinge was also deficient in *matS* binding, despite being properly folded. We propose that the interaction we assay reflects a transient 'handover' state that may be related mechanistically to MukBEF displacement from MatP-bound *matS* sites within *ter*, with the possibility that free MatP is bound to MukBEF complexes at the time of their displacement through the hinge interaction characterized here. If this is the case, then there must be other conformations or 'states' compatible with MatP-*matS* binding to MukBEF. Indeed, since the experimental work here was completed, a cryo-EM structure of MukB$^{EQ}$EF complexes with bound MatP-*matS* has demonstrated MatP-*matS* binds specifically to the MukB coiled-coils in the region of the joint when ATP hydrolysis is impaired (*Bürmann et al., 2021*). Since we are confident that the interaction of MatP with the MukB hinge characterized here is functionally relevant, given the specificity and affinity of binding with defined stoichiometry in different assays, it appears likely that the MatP hinge binding and the MatP-*matS* coiled-coil binding represent different states in the overall reaction pathway; the relationship between these two states will only be revealed by future studies. Nevertheless, we propose that the 'locked state' present in the hydrolysis-impaired MukB$^{EQ}$EF in the cryo-EM structure will be different to a hydrolysis-competent state in which we would expect MatP-*matS* to be released from the MukB coiled-coils. As is often the case, our in vitro biochemical assays using an isolated hinge domain reflect a 'snapshot' rather than complete in vivo behavior, which in the case of MukBEF involves very large multiprotein dimer of dimer complexes, whose conformations may well involve the MukB hinge associating with the ATPase heads, through bending of the coiled-coils at the elbow, close to the middle of the coiled-coil (*Bürmann et al., 2019*).

## Perspective

The work here highlights the importance of combining different biochemical assays using partially reconstituted complexes in vitro with quantitative in vivo analysis using fully functional complexes. The competition between MatP and topoIV for MukB hinge binding revealed here leads us to conclude that the characterized depletion of MukBEF from *ter*, dependent on MatP binding *matS*, may act to deplete both MukBEF and topoIV from *ter* and enrich them in the *ori* region, which is associated with the major MukBEF clusters (*Danilova et al., 2007*; *Mäkelä and Sherratt, 2020*). Cohesion time for newly replicated *ori* loci, reflecting in large part the time for decatenation, has been reported to be decreased by modestly increasing topoIV levels (*Wang et al., 2008*) and increased by impairing topoIV activity (*Zawadzki et al., 2015*). In the absence of MatP, precocious segregation of newly replicated *ter* loci has been observed (*Mercier et al., 2008*; *Nolivos et al., 2016*), consistent with increased topoIV recruitment to *ter* and enhanced *ter* decatenation in the absence of MatP. Additionally, this could lead to 'inappropriate' topoIV action within *ter* in MatP⁻ cells (knotting or supercoiling changes prior to replication and/or promiscuous catenation/premature decatenation after replication), possibly leading to the chromosome segregation defects observed in rich medium-grown MatP⁻ cells (*Espéli et al., 2012*). It is also possible that the relative depletion of topoIV from *ter* in WT cells acts to delay decatenation and therefore provide tension between segregating sister chromosomes, which might be important for coordinating segregation with cell division, as is the case with

eukaryote chromosome segregation. Therefore, the competition in MatP and topoIV binding to the MukB hinge may serve to orchestrate and coordinate MukBEF activity with the dynamics of the cell cycle. A comparable transient 'handover' state to that proposed for the MatP-MukB hinge interaction may also explain the observation that DNA and the MukB hinge compete for binding to ParC in vitro (*Vos et al., 2013b*), despite their in vivo action together requiring the participation of all three components. Indeed, one could consider that the acidic MukB hinge binding interface for MatP and ParC is effectively a DNA mimic.

Our demonstration that cells expressing the MukB^{KKK} hinge variant fail to associate with topoIV in vivo supports the observations from in vitro assays reported here and elsewhere (*Hayama et al., 2013*). Cells expressing this variant showed no major topoIV-defective phenotype as assessed by cell size distribution and only a relatively minor detectable defect in segregation of newly replicated *ori*s, intermediate between that of WT and Δ*mukB* cells. Whether this difference in *ori1* focus number between Δ*mukB* cells and *mukB*^{KKK} cells reflects a difference in decatenation frequency in these strains or is a consequence of an additional role of MukB in determining *ori* cohesion time remains to be determined. For example, it is possible that MukBEF (and MukB^{KKK}EF) might facilitate decatenation by topoIV by generating chromosome conformations that are optimal substrates for topoIV action, as has been proposed for other SMC complexes (*Charbin et al., 2014*). Our failure to observe interaction of ParC molecules with MukB^{KKK}EF complexes in cells indicates that there are no binding sites other than in the hinge for ParC/topoIV in MukBEF complexes. In contrast, the newly revealed binding site for MatP-*matS* in the MukB coiled-coils (*Bürmann et al., 2021*) likely explains why we failed to observe a MatP^- phenotype in cells expressing the MukB^{KKK} hinge mutant, whose MukB hinge domain is defective in MatP binding in vitro.

Catenation links between newly replicated sister chromosomes are believed to form as replication progresses, with topoIV being the major decatenase in *E. coli* and many other bacteria, with topoIV inactivation leading to failed chromosome segregation and cell division (*Adams et al., 1992*; *Peng and Marians, 1993*; *Zechiedrich et al., 1997*; *Wang et al., 2008*). Additionally, the type I topoisomerase, topoIII, can decatenate regions of chromosomes containing single strands, for example, at replication forks (*Lee et al., 2019*; *Nurse et al., 2003*), while FtsK-dependent XerCD site-specific recombination at the *E. coli dif* locus can efficiently remove replicative catenanes within *ter* (*Grainge et al., 2007*; *Shimokawa et al., 2013*). Loss of *E. coli* FtsK translocation activity combined with a lack of functional MukBEF leads to extensive filamentation, chromosome segregation defects, and unviability (*Sivanathan et al., 2009*), possibly because of a combined decatenation defect in such cells. We have proposed previously that some of the Muk^- phenotype, particularly the delayed segregation of newly replicated *ori*s, arises from defective decatenation by topoIV in the absence of MukBEF (*Mäkelä and Sherratt, 2020*; *Nolivos et al., 2016*; *Wang et al., 2008*). The results here, using the *mukB*^{KKK} mutant that is defective in the ParC-MukB interaction, are consistent with this, yet provide no new mechanistic insight into how the topoIV-MukBEF interaction facilitates decatenation. The substantial body of work that has characterized the MukB-topoIV interaction is relevant to reports that implicate a range of SMC complexes in acting together with type II topoisomerases (*Charbin et al., 2014*; *Piskadlo et al., 2017*; *Sen et al., 2016*; *Leonard et al., 2015*; *Kanno et al., 2015*; *Uusküla-Reimand et al., 2016*), although in the latter cases there is no direct evidence to identify the nature of any interaction. Intriguingly, *Bacillus subtilis* SMC is not known to specifically bind topoIV, but rather it does interact with the site-specific recombinase XerD, to facilitate recombination-independent expulsion of SMC complexes from *ter* (*Karaboja et al., 2021*). It is conceivable that the *B. subtilis* SMC-XerD interaction promotes decatenation by site-specific XerCD recombination at *dif* in that organism. Finally, the demonstration that SMC complexes are displaced from *B. subtilis ter* by interaction with *ter*-bound XerD (*Karaboja et al., 2021*) underlines the fact that displacement of these complexes from *ter* is not restricted to the few bacterial species that encode MukBEF-MatP and that such displacement may be a general feature of bacterial chromosome dynamics.

## Materials and methods
### Protein overexpression and purification
Two MukB hinge-based constructs were used: LH (568–863) and short hinge (SH, 667–779). SH was purified using a C-terminal 6xHis-tag, whereas LH was initially purified as an N-terminal fusion to MBP.

SH-His was overexpressed and first purified by TALON affinity chromatography as described for MukB (*Zawadzka et al., 2018*). Then, peak fractions were diluted to 100 mM NaCl and loaded onto a 5 mL HiTrap Q XL column (GE Healthcare) equilibrated in 50 mM HEPES pH 7.3, 100 mM NaCl, 1 mM EDTA, and 10% (v/v) glycerol. Elution was achieved over 40-column volumes using a gradient of 100–1000 mM NaCl. Appropriate fractions were pooled and concentrated by centrifugal filtration (Vivaspin 20, 5000 MWCO PES, Sartorius) for loading onto a Superose 6 Increase 10/300 GL (GE Healthcare) column equilibrated in storage buffer (50 mM HEPES pH 7.3, 300 mM NaCl, 1 mM EDTA, 1 mM dithiothreitol [DTT], and 10% [v/v] glycerol). Peak fractions were assessed for purity (>90%) by sodium dodecyl sulfate-polyacrylamide gel electrophoresis (SDS-PAGE; 4–20% gradient) and Coomassie staining, concentrated as appropriate by centrifugal filtration, and snap-frozen as aliquots for storage at –80 °C.

MBP-LH (and derivatives thereof) was overexpressed using the pMAL-c5X vector system according to the vendor-supplied protocol in NEBExpress cells (New England Biolabs). Glucose was present at 0.2% (w/v) throughout overexpression to repress amylase expression. For purification, cells were resuspended in lysis buffer (50 mM HEPES pH 7.3, 300 mM NaCl, 5% [v/v] glycerol) supplemented with a protease inhibitor cocktail (EDTA-free, Thermo Scientific Pierce) and mechanically lysed using a homogenizer. The lysate was clarified by centrifugation and the cleared suspension diluted fourfold in cold lysis buffer per 25 mL of extract. This was added to 5 mL (per 2 L of culture) amylose resin (New England Biolabs) equilibrated in lysis buffer and left on a tube roller shaker at 4 °C for 1 hr. This suspension was loaded onto a gravity-flow column. The settled resin was washed with 10-column volumes of lysis buffer. MBP-LH was eluted in two-column volumes using lysis buffer supplemented with 10 mM maltose. MBP was cleaved from LH using Factor Xa (New England Biolabs) at a w/w ratio of 1% Factor Xa:LH. Efficient cleavage typically required incubation at 4 °C for 36 hr. The cleaved sample was diluted until a final (NaCl) of 100 mM and then loaded onto a 5 mL HiTrap Q XL column equilibrated in 50 mM HEPES pH 7.3, 1 mM EDTA, 10 % (v/v) glycerol with 100 mM NaCl. Elution was achieved with a linear 40-column volume gradient to 1 M NaCl to isolate tagless LH and remove Factor Xa. Finally, appropriate fractions were pooled for further purification and buffer exchange by SEC as described for aforementioned for MukB.

Purification of full-length ParC and MatP (MatPΔ18C;1–132) used an N-terminal and C-terminal 6xHis-tag, respectively. Overexpression and initial purification was completed as previously published (*Nolivos et al., 2016*) with the addition of a SEC step as described above. Note that these proteins are unstable during purification when using buffers with a NaCl concentration below 300 mM. ParC R705E R729A and MatP K71E/A, Q72E/A, R75E/A, R77E/A variants were produced by site-directed mutagenesis (Q5 Site-Directed Mutagenesis Kit, New England Biolabs), verified by sequencing, and purified using the same protocol.

ParC$^{CTD}$ used an N-terminal MBP fusion for ease of purification and to improve stability of the construct as reported by *Vos et al., 2013b* and *Vos et al., 2013a*. MBP-ParC$^{CTD}$ was overexpressed and initially purified using amylose affinity chromatography as described for MBP-LH; however, the MBP fusion was not removed. Instead, MBP-ParC$^{CTD}$ was loaded onto a Superose 6 Increase 10/300 GL column (GE Healthcare) equilibrated in storage buffer.

ParE-His was overexpressed and initially purified using TALON affinity chromatography as for MukB (*Zawadzka et al., 2018*). The eluate was diluted to 100 mM NaCl and loaded onto a 1 mL HiTrap Q XL column equilibrated in 50 mM HEPES pH 7.3, 100 mM NaCl, 1 mM EDTA, and 10% (v/v) glycerol. Elution was achieved over a 20-column volume gradient to 1 M NaCl. Selected fractions were passed over a Superose 6 Increase 10/300 GL (GE Healthcare) column equilibrated in storage buffer. TopoIV reconstitution required mixing of equimolar amounts of ParC and ParE, which were incubated for 30 min on ice. Efficient reconstitution under these conditions was verified in analytical SEC. All protein concentrations stated correspond to the protein as a monomer unless indicated otherwise.

## Fluorophore labeling of MatP and MukB

Endogenous MatP contains no native cysteines; therefore for introduction of a fluorophore, a single cysteine was engineered into the C-terminal linker between MatP and its 6xHis-tag (SDPNSSSVDKL CAAALEHHHHHH). Immediately following purification, MatP-Cys-His was treated with 0.2 mM TCEP for 30 min at 22°C ± 1 °C. Cy3B maleimide was dissolved in anhydrous DMSO to produce a 10 mM stock and immediately added to MatP-Cys-His at a sixfold molar excess. This reaction was rotated

end-over-end at 4 °C for 16 hr whilst protected from light and then quenched by addition of DTT to a final concentration of 5 mM. Excess dye was removed by SEC. Labeling efficiency was calculated (89%) by spectrophotometry and using a vendor-supplied correction factor of 0.08 for Cy3B absorbance at 280 nm.

Unnatural amino acid labeling was used for conjugating fluorophores to MukB. S718 was mutated to an amber stop codon in a pBAD24 expression vector. This was co-transformed with pEVOL-pAzF (Addgene) into an *E. coli* C321.ΔA strain (*Lajoie et al., 2013*; *Chin et al., 2002*), where endogenous MukB has a C-terminal 3xFLAG tag, and UAG has been reassigned as a sense codon (FW01). 1% (w/v) glucose was present throughout overexpression, which was induced by the addition of L-arabinose to 0.4% (w/v) and p-azidophenylalanine (azF) to 1 mM at an $OD_{600}$ of 0.6. Expression proceeded for 4 hr at 30 °C. azF-MukB was purified as for the wild-type protein (*Zawadzka et al., 2018*) with an additional step post-TALON resin, where the eluate was incubated with 125 µL of equilibrated ANTI-FLAG M2 agarose affinity gel (Sigma) for 1 hr on a rolling shaker at 4 °C before being poured onto a column. The flow-through was recovered and processed as for wild-type protein. A 20-fold excess of the dye (DBCO-TAMRA) was added to MukB at ~15 µM. The reaction was left to proceed for 1 hr at 22°C ± 1 °C and then moved to 4 °C for 16 hr in the absence of light. Free dye was removed from labeled MukB by SEC using a Superdex 200 10/300 GL column (GE Healthcare). Labeling efficiency (typically 40–70%) was calculated using the molar extinction coefficient of the TAMRA dye at 547 nm ($92,000\ M^{-1}\ cm^{-1}$), and a 0.3 correction factor for absorption at 280 nm by the dye. The suitability of substitution of S718 to a phenylalanine analogue was verified in vitro by measuring the ATPase activity of azF-MukB (*Figure 3—figure supplement 1B*) and in vivo by assaying its ability (as S718F) to rescue the temperature-sensitive growth defect of a ΔmukB strain as previously described (*Zawadzka et al., 2018*; *Figure 3—figure supplement 1C*).

## DNA preparation

For nMS and native PAGE, 50 bp *matS*-containing or nonspecific double-stranded DNA of the same size and GC content was prepared by slowly annealing complementary oligonucleotides: 50 bp *matS* oligo 1 5′ CAG AGT TAA TCA GAA CGG TGA CAA TGT CAC AAA GAA AAA GAA CCT GTG CG 3′; 50 bp *matS* oligo 2 5′ CGC ACA GGT TCT TTT TCT TTG TGA CAT TGT CAC CGT TCT GAT TAA CTC TG 3′; 50 bp nonspecific oligo 1 5′ CAG AGT TAA TCA CAA CGG TTC TCG ATC ATC AAA GAA AAA CAA GCT GTG CG 3′ and 50 bp nonspecific oligo 2 5′ CGC ACA GCT TGT TTTT CTT TGA TGA TCG AGA ACC GTT GTG ATT AAC TCT G 3′. All oligos were dissolved in annealing buffer (10 mM Tris-HCl pH 8.0, 10 mM NaCl, and 1 mM EDTA), mixed at an equimolar ratio, heated at 95 °C for 5 min and cooled slowly in 0.1 °C increments to 10 °C over 6 h. Double-stranded DNA formation was assessed by agarose gel electrophoresis.

For FCS, 15 bp *matS*-containing or nonspecific DNA hairpins were produced by resuspending the following oligonucleotides in annealing buffer to produce 50 µM stocks: 5′ GTG ACA ATG TCA C TTC CCT G TGA CAT TGT CAC 3′ and 5′ GTT CTC GAT CAT C TTC CCT G ATG ATC GAG AAC 3′ for *matS* and nonspecific DNA, respectively. Both DNAs were heated at 95 °C for 15 min and then immediately placed in an ice-water bath for 10 min. Selective hairpin formation was assessed by PAGE using 15% TBE (pH 7.4) gels later stained with 0.5 µg/mL ethidium bromide 1× TBE solution.

## Circular dichroism (CD) spectroscopy

CD spectra were collected at 20 °C using a 0.1 cm quartz cuvette in a JASCO J-815 spectropolarimeter equipped with a JASCO CDF-426S Peltier temperature controller. 0.05–0.1 mg/mL samples were buffer exchanged into 10 mM phosphate buffer pH 8.0 and 20 mM NaCl. Data was acquired across a 190–250 nm absorbance scan using a band width of 1 nm, a data pitch of 0.1 nm, and scan rate of 100 nm/min. Nine scans were accumulated and averaged (technical repeats), and the data normalized to molar ellipticity by calculation of the cell path length and concentration of peptide bonds. A buffer-only baseline was subtracted from all data. Final data is from the measurement of a single sample preparation.

**Table 1.** Theoretical masses of proteins and DNA used in native mass spectrometry (nMS).
Predicted and measured masses of MukBEF, topoIV, and MatP components or variants and also DNA substrates. Errors are the standard deviation in mass determination.

| Protein | Theoretical mass of monomer (Da) * | Measured oligomeric state(s) | Theoretical mass(es) of native state(s) (Da) | Measured mass(es) (Da) |
|---|---|---|---|---|
| LH (MBP cleaved) | 34334.87 | Dimer | 68669.74 | 68668 ± 1 |
| SH-His | 15204.73 | Monomer/dimer mix † | 15204.73/ 30409.46 | 15105 ± 5 30188 ± 12 |
| MatP (i.e., MatPΔ18C-His) | 18329.85 | Dimer | 36659.7 | 36418 ± 1 |
| MatP K71E, Q72E, R75E, R77E | 18277.63 | Dimer | 36555.26 | 36429 ± 15 |
| His-ParC | 86152.75 | Dimer | 172305.5 | 172076 ± 2 |
| His-ParC R705E R729A | 86040.57 | Dimer | 172081.14 | 171863 ± 3 |
| MBP-ParC$^{CTD}$ | 73422.74 | Monomer | 73422.74 | 70761 ± 6‡ 71447 ± 8‡ |
| ParE-His | 72484.36 | Dimer | 144968.72 | 144717 ± 2 |
| 50 bp *matS* DNA | 30925.10 | N/A | 30766.82 | 30801 ± 14 |
| 50 bp *nonspecific* DNA | 30925.10 | N/A | 30766.82 | 30780 ± 15 |

*Masses include first methionine.
†From this work.
‡Degradation products.

## Isothermal titration calorimetry (ITC)

Binding was assayed in a Malvern PEAQ ITC instrument at 25 °C. Data were analyzed and fitted to a single binding site model using the manufacturer's software. Means and standard deviations of the obtained parameters were derived from triplicate experimental repeats.

## Native mass spectrometry (NMS)

Prior to nMS analysis, individual proteins were buffer exchanged into 200 mM ammonium acetate pH 8.0 either by SEC or using Biospin-6 (BioRad) columns and introduced directly into the mass spectrometer using gold-coated capillary needles (prepared in-house). To reconstitute the complexes, buffer-exchanged proteins were mixed in different ratios and incubated on ice for 10 min. Data were collected on a Q-Exactive UHMR mass spectrometer (ThermoFisher). The instrument parameters were as follows: capillary voltage 1.1 kV, quadrupole selection from 1000 to 20,000 m/z range, S-lens RF 100%, collisional activation in the HCD cell 50–200 V, trapping gas pressure setting kept at 7.5, temperature 100–200 °C, and resolution of the instrument 12,500. The noise level was set at 3 rather than the default value of 4.64. No in-source dissociation was applied. Data were analyzed using Xcalibur 4.2 (Thermo Scientific) and UniDec (*Marty et al., 2015*). The theoretical and measured masses of all constructs used in nMS experiments in this study are listed in *Table 1*. At least three biological repeats of data collection for all spectra were completed.

## Analytical ultracentrifugation (AUC)

All AUC data were obtained on a Beckman XL-I using absorbance optics. MatP and LH were taken at 100 µM in 50 mM HEPES pH 7.5, 100 mM NaCl, and 1 mM MgCl$_2$. Sedimentation velocity experiments were carried out at 40,000 rpm using an AnTi60 rotor at 20 °C. Cells were scanned every 10 min at 280 nm. All data were analyzed using SEDFIT (*Schuck, 2000*). Presented data is from the measurement of a single sample.

## Fluorescence correlation spectroscopy (FCS)

FCS experiments were performed using a bespoke confocal microscope with continuous excitation at 532 nm (50 µW, Samba, Cobolt). Time traces were acquired for 30 s using a SPQR14 avalanche

photodiode (PerkinElmer), and autocorrelation functions were produced in real time using a Flex02-02D correlation card (Correlator.com). Data acquisitions were performed with custom software written in LabVIEW (National Instruments; RRID:SCR_014325). Fluorescence arrival times were recorded on a SPQR-14 detector (PerkinElmer) and processed using custom software written in LabVIEW, MATLAB (MathWorks; RRID:SCR_001622), and Python (RRID:SCR_008394).

Samples of 1 nM fluorophore-labeled protein were deposited onto PEGylated slides in FCS buffer (20 mM Tris-HCl pH 7.5, 50 mM NaCl, 0.3 mg/mL BSA, 2 mM DTT, 0.05% [v/v] Tween-20). All buffers were UV-bleached before use. The diffusion times of Cy3B-labeled MatP and TMR-labeled MukB (conjugated at S718$^{TAG}$ using UAA as described above) alone were established (*Figure 3—figure supplement 1A*) to allow for identification of the MukB-bound MatP population. Complex samples with more than one component were incubated for 10 min at 22 °C ± 1 °C prior to data acquisition. Competitor proteins were added at 2.5–50× molar excess. Data for each sample was collected from >20 datasets (inclusive of both biological and technical repeats).

For the single diffusing species, the autocorrelation function $G(\tau)$ is given by

$$G(\tau) = A_0 + \frac{1}{n} \frac{1}{\left(1+\frac{\tau}{\tau_1}\right)} \frac{1}{\sqrt{1+\frac{\tau}{SP^2\tau_1}}} \left(1 + \frac{Te^{-\frac{\tau}{\tau_{trip}}}}{1-T}\right)$$

where $A_0$ is the offset, *n* is the effective number of particles in the confocal volume, *SP* is the structural parameter that describes elongation of the confocal volume, *T* is the fraction of MatP in triplet state, $\tau_1$ is the characteristic diffusion time of free MatP, and $\tau_{trip}$ is the characteristic residence time in triplet state.

For a two-component system, such as that consisting of free MatP and MukB-bound MatP, the correlation function becomes

$$G(\tau) = A_0 + \frac{1}{n(F+\alpha(1-F))^2} \left(1 + \frac{Te^{-\frac{\tau}{\tau_{trip}}}}{1-T}\right) \times \left[\frac{F}{(1+\tau/\tau_1)} \frac{1}{\sqrt{1+\tau/(SP^2\tau_1)}} + \alpha^2 \frac{1-F}{(1+\tau/\tau_2)} \frac{1}{\sqrt{1+\tau/(SP^2\tau_2)}}\right]$$

where $A_0$, *n*, *SP*, *T*, and $\tau_{trip}$ are the same parameters as described above. $\tau_1$ is the characteristic diffusion time of free MatP, and $\tau_2$ is the characteristic diffusion time of bound-MatP (MukB-MatP). *F* is the fraction of molecules of MatP, and $\alpha$ is the relative molecular brightness of MatP and MukB-MatP (regarded as 1).

FCS data were fitted using PyCorrFit software (*Müller et al., 2014*). A data range of 300–750 was used for channels setting that defined the timescale as $1.8000 \times 10^{-2}$ to $3.0802 \times 10^{+2}$ ms. The following constraints were set for fitting: $\tau_{trip}$ at 100–1000 μs and $\tau_1$ at 0–100 ms. For the single diffusing species fitting (MatP-Cy3B), *SP* was defined as the fixed parameter at default value of 5. For the two-component system fitting, $\tau_1$ was set as 4.5 ms for free MatP-Cy3B (diffusion time of ~4.5 ms). *SP* was the fixed parameter with a default value of 5, $\tau_{trip}$ was 100–1000 μ, and $\tau_2$ as 20–100 ms. Binding data were fitted using the Hill equation with Origin software (version 2017, OriginLab Corporation).

## Native polyacrylamide gel electrophoresis (PAGE)

Samples were prepared in 50 mM HEPES pH 7.5, 100 mM NaCl, 1 mM MgCl$_2$, 1 mM DTT, and 10% glycerol (v/v) buffer. In experiments used to monitor LH/LH$^{KKK}$ binding to MatP and ParC, LH (3 μM final) was mixed with ParC and/or MatP at a 1:2(:2) molar ratio for 30 min on ice. For experiments assaying binding of MatP, or variants, to DNA and/or LH MukB, MatP (7 μM final) was mixed with LH and/or DNA in a 1:1(:1) molar ratio and incubated on ice for 30 min. Duplicate 10 μL samples were loaded onto 10% native polyacrylamide gels poured in 125 mM Tris-HCl, pH 8.8. Following electrophoresis, gels were stained with Coomassie blue or ethidium bromide. Gels are representative of at least two biological replicate experiments.

## Analytical size-exclusion chromatography (SEC)

For assessing complex formation, proteins were mixed at the indicated ratios and equilibrated in 50 mM HEPES pH 7.5, 100 mM NaCl, 1 mM MgCl$_2$, 1 mM DTT, and 10 % (v/v) glycerol buffer for 1 h on ice. 100 μL of these mixtures (containing <900 μg of total protein) were loaded onto a Superose 6 Increase 10/300 column equilibrated in the same buffer. Separation was conducted at a flow rate of

0.5 mL/min and 0.3 mL or 0.5 mL fractions collected for SDS-PAGE analysis. Final data presented is from a single experiment.

## ATP hydrolysis assays

An EnzCheck Phosphate Assay Kit (ThermoFisher) was used as described previously (*Zawadzka et al., 2018*), with the exception that all final reactions contained 65 mM NaCl. The reaction was started by addition of ATP to a final concentration of 1.3 mM. MukE and MukF were purified as described (*Zawadzka et al., 2018*). Datasets comprised three biological replicate experiments.

## Functional analyses in vivo

The ability of MukB variants to complement the temperature-sensitive growth defect of a Δ*mukB* strain was tested as described previously (*Zawadzka et al., 2018*). Plate images for MukB^KKK are representative of two biological replicate experiments. Plate images for MukB^S718F show a single experiment (*Figure 3—figure supplement 1C*).

## Epifluorescence microscopy and PALM

The conditions for all imaging and analysis are as described in *Mäkelä and Sherratt, 2020*; *Zawadzki et al., 2015*. Briefly, single-cell parameters including cell dimensions, spot localization, and pixel intensity values were extracted using SuperSegger (*Stylianidou et al., 2016*; RRID:SCR_018532) in MATLAB (MathWorks; RRID:SCR_001622). For fluorescence intensity profiles as a function of *ori1*/*ter3* distance, cell pixel intensities were normalized by subtracting the average cell intensity and dividing by the maximum intensity. The distance from each pixel to the closest *ori1* and *ter3* markers was measured and the average intensity from a population of cells as a function of distance was estimated. All custom scripts used in MATLAB are attached as source code files. The analyses of single-molecule ParC cellular diffusion and how it is impacted by the presence of MukB were as in *Zawadzki et al., 2015*. Cells for imaging were grown in M9 glycerol minimal media at 30 °C. The genotypes of all strains used are described and/or cited in *Figure 5* and *Figure 5—figure supplement 1*. Datasets are derived from three biological replicate experiments.

# Acknowledgements

We thank all members of the Sherratt lab for useful discussions and David Staunton for expert technical assistance with ITC. Achillefs Kapanidis (Dept. of Physics, University of Oxford, UK) provided facilities for FCS analysis. We thank Madhu Srinivasan (Dept. of Biochemistry, University of Oxford, UK), Frank Bürmann and Jan Löwe (MRC LMB, Cambridge, UK) for helpful discussions. We also thank K Zawadzka and P Zawadzki (MNM, Poznan, Poland), whose actions and ideas helped initiate the work in this paper.

# Additional information

## Funding

| Funder | Grant reference number | Author |
|---|---|---|
| Wellcome Trust | 200782/Z/16/Z | David J Sherratt |
| Medical Research Council | MR/N020413/1 | Carol V Robinson |

The funders had no role in study design, data collection and interpretation, or the decision to submit the work for publication.

## Author contributions

Gemma LM Fisher, Conceptualization, Data curation, Formal analysis, Methodology, Resources, Writing – original draft, Writing – review and editing; Jani R Bolla, Man Zhou, Data curation, Formal analysis, Investigation, Methodology, Writing – review and editing; Karthik V Rajasekar, Conceptualization, Data curation, Formal analysis, Investigation, Methodology, Resources, Writing – review and editing; Jarno Mäkelä, Formal analysis, Investigation, Methodology, Software, Writing – review and

editing; Rachel Baker, Data curation, Formal analysis, Investigation, Resources, Writing – review and editing; Josh P Prince, Data curation, Formal analysis, Investigation, Writing – review and editing; Mathew Stracy, Data curation, Formal analysis, Investigation, Methodology, Software, Writing – review and editing; Carol V Robinson, Formal analysis, Funding acquisition, Methodology, Supervision, Writing – review and editing; Lidia K Arciszewska, Conceptualization, Formal analysis, Investigation, Methodology, Supervision, Writing – original draft, Writing – review and editing; David J Sherratt, Conceptualization, Formal analysis, Funding acquisition, Investigation, Methodology, Supervision, Writing – original draft, Writing – review and editing

### Author ORCIDs
Gemma LM Fisher ![ORCID] http://orcid.org/0000-0001-8468-5032
Jani R Bolla ![ORCID] http://orcid.org/0000-0003-4346-182X
Karthik V Rajasekar ![ORCID] http://orcid.org/0000-0002-8146-6560
Josh P Prince ![ORCID] http://orcid.org/0000-0003-0877-7538
Lidia K Arciszewska ![ORCID] http://orcid.org/0000-0002-0252-4874
David J Sherratt ![ORCID] http://orcid.org/0000-0002-2104-5430

### Decision letter and Author response
Decision letter https://doi.org/10.7554/eLife.70444.sa1
Author response https://doi.org/10.7554/eLife.70444.sa2

## Additional files

### Supplementary files
• Transparent reporting form

• Source code 1. All custom MATLAB scripts used for analysis of epifluorescence microscopy data in *Figure 5* and *Figure 5—figure supplement 1*.

### Data availability
Source data files have been provided for all gel-based figures. Source data files for other assays have been included where indicated. All custom MATLAB scripts are provided as Source Code 1. All reagents are available upon reasonable request.

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
