## [Decision Letter]

**Acceptance summary:**

This paper is of potential interest to an audience of biochemists, cell biologists, and structural biologists working in the area of bacterial chromosome organization and segregation. A range of elegant in vitro and in vivo analyses is used to provide compelling biochemical evidence for the interaction of MatP and ParEC at the hinge of MukB, the condensin of Enterobacteria. The model for competitive interactions as a means of spatiotemporal regulation of MukBEF-topoisomerase IV activities is discussed.

**Decision letter after peer review:**

Thank you for submitting your article "Competitive binding of MatP and topoisomerase IV to the MukB hinge modulates chromosome organization-segregation." for consideration by *eLife*. Your article has been reviewed by 3 peer reviewers, and the evaluation has been overseen by a Reviewing Editor and Jessica Tyler as the Senior Editor. The reviewers have opted to remain anonymous.

Essential revisions:

While the reviewers acknowledged the elegance and quality of the in vitro data, significant concerns were raised regarding the interpretation and significance of the described interactions in vivo.

1. The major conclusions about the interaction between MatP and MukBEF require additional support (see reviewers' individual comments). The significance of the absence of interaction should be documented using at least one of the methods that can reveal the inhibition/displacement of MukBEF by MatP, either by SIM (Mäkela and Sherratt, 2020), by ChIP seq of MukBKKK (as in Nolivos et al., 2015), or by Chromosome-Conformation-Capture (Lioy et al., 2018).

2. The results obtained should be discussed in light of results obtained by Bürmann and colleagues (BioRXiv, 2021)

*Reviewer #1 (Recommendations for the authors):*

I have a few points to raise on the manuscript:

– line 46-47: The sentence 'MukBEF act to position the origin of replication' is misleading. It suggests the direct involvement of MukBEF in the positioning. But it could also be a consequence of its role in the management of the chromosome. Note that the results of Figure 5 suggest that ori are well positioned in a MukBKKK mutant, which seems impaired in ParC and MatP binding.

– line 64-65: while reading this paper, I looked for what was known about MukBEF in the literature. I found a very interesting article in BioXiv by the team of Jan Löwe, in which an interaction between MatP and the joint of one MukB protein and interaction between MatP and MukE are described (CryoEM+ in vivo mapping of DNA topology). It raises many interesting possibilities that should be discussed. It might explain how MukB binds to MatP when bound to matS sites whereas te MukB hinge and matS compete for binding to MatP (line 209, line 373-394).

– line 248-248: the title of the section is badly worded. I guess MuK^+^ refers to the fact that MukBKKK cells grow 'normally' (Figure 5C). Ori positioning seems fairly normal (Figure 5B). However, the MukBKKK signal is clearly diffuse (Figure 5A), which makes dubious the conclusions made about (Figure 5D and 5E). Could the authors had ChIP MukBKKK to show whether there are still peaks at matS sites?

– line 311-314: attenuate the sentence: MukBEF is supposed to position.

*Reviewer #2 (Recommendations for the authors):*

At this stage, I believe this paper should not be considered for publication in *eLife* but has to be considerably revised to address the points raised in the public review section.

*Reviewer #3 (Recommendations for the authors):*

1. A model figure would assist in illustrating the discussion in the "TopoIV interaction with the MukB hinge" Discussion section.

2. Is there any understanding of why MukE foci are more diffuse in MukBKKK cells in comparison to MukB WT cells? Is this mutant defective in DNA binding, or is there another possible impairment?

3. It would help to explain more clearly in the main text where MatP dimers elute in analytical SEC (Figure 2A and 3C). I was confused until I found the explanation in the Figure 3 caption.

4. The language on about MukBEF localization in ΔmatP cells (p. 7 line 262) is a little confusing – "positioned equally distant for ori1 and ter3" makes it sound as if MukBEF is not associated with either ori or ter, whereas it appears to colocalize with both. Rewording this sentence might clarify the point that the MukBEF remains colocalized with ori but is now equally colocalized with ter in ΔmatP cells.

5. The labeling of complexes in the native mass spectra is a little confusing in that the subunit stoichiometries are omitted in some cases and included in others. Although I understand that this convention is probably being used to match the convention in the text (e.g., LH-MatP instead of LH2-MatP2), perhaps there is a way to make the exact stoichiometries of the detected complexes clearer.

6. The fluorescence imaging methods are not described in this manuscript; instead previous papers from the corresponding author's lab are cited. Both previous papers are open access, and so this does not represent a major impediment to the reader in finding a description of the methods, but I mention it in case there is an *eLife* policy against citing previous papers instead of describing all methods in the manuscript itself.

---

## [Author Response]

Essential revisions:While the reviewers acknowledged the elegance and quality of the in vitro data, significant concerns were raised regarding the interpretation and significance of the described interactions in vivo.1. The major conclusions about the interaction between MatP and MukBEF require additional support (see reviewers' individual comments). The significance of the absence of interaction should be documented using at least one of the methods that can reveal the inhibition/displacement of MukBEF by MatP, either by SIM (Mäkela and Sherratt, 2020), by ChIP seq of MukBKKK (as in Nolivos et al., 2015), or by Chromosome-Conformation-Capture (Lioy et al., 2018).2. The results obtained should be discussed in light of results obtained by Bürmann and colleagues (BioRXiv, 2021)

Thank you for the support and positive response.

Regarding point 1: In the absence of having the personnel, laboratory space and capacity to make new reagents following David Sherratt’s retirement and dis-banding of the research group earlier this year, we completed further extensive quantitative analyses of the in vivo data previously acquired – see revised Figure 5 and the accompanying new Supplementary Figure. The opportunity to complete a more detailed analyses has been fruitful and led us to conclude that MukB^KKK^EF expressing cells have a substantial defect in an uncharacterized MukBEF function whilst remaining sufficiently functional to form some chromosome-associated clusters and to render cells temperature-resistant in growth. Importantly, the new analyses did not reveal an in vivo defect in interaction of MukB^KKK^EF complexes with MatP-*matS*. This is perhaps not surprising in light of the demonstration by Bürmann *et al.,* that there is an additional binding site for MatP-*matS* in the MukB coiled-coils. This revelation also makes the proposed additional experiments rather redundant (unless a new project was initiated in which the MukB coiled-coil binding site for MatP-*matS* was ablated; we hope that Bürmann and colleagues will undertake this new project). Moreover, it is our understanding that *eLife* policy is not to request further experiments that represent a new line of approach.

Regarding point 2, we now discuss the results of Bürmann et al., within the “The MukB hinge and matS DNA compete for binding to MatP” Discussion section (starting line 431) of the manuscript. Their presented structures and models are of considerable importance (and excitement!) for the field and have helped us to better understand some of our own results and strengthen our view that the MatP-MukB hinge complexes we analyzed represent an intermediate ‘handover’ state. It appears likely that our MatP binding to the hinge domain and MatP-matS coiled-coil binding represent different states in the overall reaction pathway. Currently, we have insufficient information to present a coherent model detailing the relationship between these two states including the presumed conformational transitions that must occur; this must be revealed by future studies.

Reviewer #1 (Recommendations for the authors):I have a few points to raise on the manuscript:– line 46-47: The sentence 'MukBEF act to position the origin of replication' is misleading. It suggests the direct involvement of MukBEF in the positioning. But it could also be a consequence of its role in the management of the chromosome. Note that the results of Figure 5 suggest that ori are well positioned in a MukBKKK mutant, which seems impaired in ParC and MatP binding.

Thank you! We have revised and clarified this in the revised manuscript.

– line 64-65: while reading this paper, I looked for what was known about MukBEF in the literature. I found a very interesting article in BioXiv by the team of Jan Löwe, in which an interaction between MatP and the joint of one MukB protein and interaction between MatP and MukE are described (CryoEM+ in vivo mapping of DNA topology). It raises many interesting possibilities that should be discussed. It might explain how MukB binds to MatP when bound to matS sites whereas te MukB hinge and matS compete for binding to MatP (line 209, line 373-394).

We were aware of this work after our own was completed, but could not initially cite it because it had not been put on bioRxiv. Our revised Discussion now discusses this work in terms of our ideas about ‘hand-on’ models.

– line 248-248: the title of the section is badly worded. I guess MuK^+^ refers to the fact that MukBKKK cells grow 'normally' (Figure 5C). Ori positioning seems fairly normal (Figure 5B). However, the MukBKKK signal is clearly diffuse (Figure 5A), which makes dubious the conclusions made about (Figure 5D and 5E). Could the authors had ChIP MukBKKK to show whether there are still peaks at matS sites?

We have clarified the text. We neither have nor have the means to retrospectively acquire the suggested ChIP data following the retirement of Sherratt and the subsequent dispersal of his laboratory. It was always our understanding that *eLife* should not ask for further new areas of experimentation (this was certainly the case in the past). As we describe in detail above, we are of the opinion that even calibrated ChIP-seq data would not provide insightful information on the mutant protein interaction with chromosomes, given the results of our new analyses. Finally, the Bürmann structure, showing a second binding site for MatP, leads us to believe that the hinge mutant that we characterize is still likely to bind MatP-*matS*.

– line 311-314: attenuate the sentence: MukBEF is supposed to position.

Attended to.

Reviewer #3 (Recommendations for the authors):1. A model figure would assist in illustrating the discussion in the "TopoIV interaction with the MukB hinge" Discussion section.

We do not believe a ‘model Figure’ would help complement or clarify the description of the “TopoIV interaction with MukB hinge” section of the Discussion and instead now refer the reader to the schematics of Figure 1A to appreciate the potentially flexible positioning of the C-terminal domains of ParC – lines 400-403. Our suggestions for conformational transitions within both/either ParC and MukB for DNA exit and entry are speculative at this stage only and hopefully will stimulate future work by other groups in the field.

2. Is there any understanding of why MukE foci are more diffuse in MukBKKK cells in comparison to MukB WT cells? Is this mutant defective in DNA binding, or is there another possible impairment?

The diffusive nature of MukB^KKK^EF foci indicates impaired chromosome association associated with impaired overall MukBEF function, consistent with the reviewer’s suggestion. Parenthetically, in unpublished work we have observed that MukB mutants with a modest defect in in vitro ATP hydrolysis also can give rather diffuse foci and we suspect that other types of mutants e.g., in DNA binding or in elbow flexibility, may give a similar phenotype, but note that head mutants that fail to bind DNA in vitro e.g., equivalent to those reported by Woo *et al.,* for *Haemophilus,* are temperature-sensitive and do not form foci (in the PhD thesis of F. Wagner). Our intuition is that hypomorphic DNA binding mutants could give a similar phenotype to that revealed here. Single-molecule tracking by PALM would have established the chromosome on-off rates of these mutant complexes, but any increase in off rate could result from many different defects in the overall cycle of DNA binding and dissociation and accompanying ATP binding and hydrolysis.

3. It would help to explain more clearly in the main text where MatP dimers elute in analytical SEC (Figure 2A and 3C). I was confused until I found the explanation in the Figure 3 caption.

We thank the reviewer for this helpful suggestion. This is now integrated into the main text – now lines 158-160.

4. The language on about MukBEF localization in ΔmatP cells (p. 7 line 262) is a little confusing – "positioned equally distant for ori1 and ter3" makes it sound as if MukBEF is not associated with either ori or ter, whereas it appears to colocalize with both. Rewording this sentence might clarify the point that the MukBEF remains colocalized with ori but is now equally colocalized with ter in ΔmatP cells.

We again thank the reviewer for highlighting this. We have clarified this point in the main text – now lines 322-325.

5. The labeling of complexes in the native mass spectra is a little confusing in that the subunit stoichiometries are omitted in some cases and included in others. Although I understand that this convention is probably being used to match the convention in the text (e.g., LH-MatP instead of LH2-MatP2), perhaps there is a way to make the exact stoichiometries of the detected complexes clearer.

Indeed, this was our logic. We have now annotated all mass spectra, and other datasets where appropriate, with the complex stoichiometry and reflected this in the main text when describing complex formation.

6. The fluorescence imaging methods are not described in this manuscript; instead previous papers from the corresponding author's lab are cited. Both previous papers are open access, and so this does not represent a major impediment to the reader in finding a description of the methods, but I mention it in case there is an eLife policy against citing previous papers instead of describing all methods in the manuscript itself.

More detailed methods have now been added.